# DiffuPhyGS: Text-to-Video Generation with 3D Gaussians and Learnable Physical Properties via Diffusion Priors

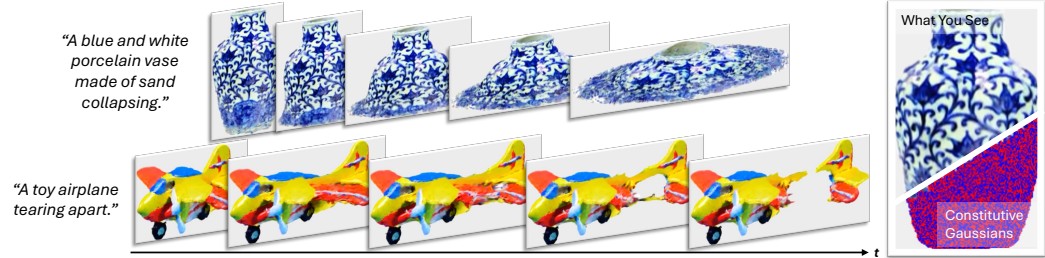

Figure 1: *DiffuPhyGS* is a text-to-3D dynamics pipeline that generates detailed 3D objects with physics-grounded and 3D geometry-aware motion

## ABSTRACT

Generating realistic 3D object videos is crucial for virtual reality and digital content creation. However, existing 3D dynamics generation methods often struggle to achieve high-quality appearance and physics-aware motion, relying on manual inputs and pre-existing models. To address these challenges, we propose Diffu-PhyGS, a novel framework that generates high-quality 3D objects with realistic and learnable physical motion directly from text prompts. Our approach features an LLM-Chain-of-Thought-based Iterative Prompt Refinement (LLM-CoT-IPR) method, which obtains prompt-aligned 2D and multi-view 3D diffusion priors to guide Gaussian Splatting (GS) to generate 3D objects. We further enhance 3D generation quality with a Densification-by-Adaptive-Splitting (DAS) mechanism. Next, we employ a material property decoder that utilizes a Mixture-of-Experts Material Constitutive Models (MoEMCMs) to predict the mixed material properties of the 3D object. We then apply the Material Point Method (MPM) to deform 3D Gaussian kernels, ensuring physics-grounded motion guided by implicit and explicit physical priors from the video diffusion model and a velocity loss function. Extensive experiments show DiffuPhyGS outperforms other methods in generating realistic physics-grounded motion across diverse materials.

## 1 INTRODUCTION

Generating realistic, physics-grounded 3D motion from text is valuable for applications in virtual reality, video games, animation, and robotics. In contrast to general 2D video generation methods (e.g., diffusion-based models Wu et al. (2025); Wan et al. (2025)) that implicitly learn motion generation from video data and produce only image-space sequences without explicit 3D geometry, materials, or controllable physical forces, 3D object-based motion generation offers stronger physical grounding and intrinsic 3D spatial awareness, resulting in more realistic and physically consistent results Huang et al. (2024); Lin et al. (2025); Zhang et al. (2024b); Liu et al. (2024). Recent advances in text-to-3D generation have enabled the creation of 3D assets Chen et al. (2024b); Tang et al. (2023a); Poole et al. (2022); Lin et al. (2023); Liang et al. (2024b); Yi et al. (2024), paving the way for incorporating dynamic behaviors. However, these text-to-3D methods often suffer from

prompt misalignment, viewpoint inconsistency, and limited visual quality. Moreover, existing 3D object-based dynamics generation approaches frequently fail to achieve both high-quality appearance and physics-aware motion, often relying on manual inputs and pre-existing 3D object models, which is inefficient and inconvenient. For example, PhysGaussian Xie et al. (2024) integrates physical simulations into 3D Gaussians but requires pre-existing 3D object models and manual material specification, which is time-consuming and requires expertise. Similarly, approaches like Dream-Physics Huang et al. (2024), OmniPhyGS Lin et al. (2025), PhysDreamer Zhang et al. (2024b), and Physics3D Liu et al. (2024) use video diffusion models for guidance but lack explicit physical constraints, leading to artifacts and poor generalization.

Prior works in the domain Huang et al. (2024); Lin et al. (2025); Zhang et al. (2024b); Liu et al. (2024) leverage video diffusion models for plausible 3D motions, prioritizing perceptual realism over physical accuracy, often resulting in unnatural deformations or velocities. They also do not fully integrate high-fidelity 3D appearance with physics-aware dynamics in a text-to-motion pipeline, nor handle complex material variations within objects.

To address these challenges, we introduce *DiffuPhyGS*, a framework for generating physics-grounded 3D object videos from text. To mitigate prompt misalignment in text-to-3D generation Poole et al. (2022); Lin et al. (2023), we propose LLM-Chain-of-Thought-based Iterative Prompt Refinement (LLM-CoT-IPR) for iterative refinement. We employ multi-view 3D diffusion priors to reduce viewpoint inconsistencies and Densification-by-Adaptive-Splitting (DAS) in Gaussian Splatting to capture fine details of the 3D object. For motion, unlike methods relying solely on implicit guidance Huang et al. (2024); Lin et al. (2025); Zhang et al. (2024b); Liu et al. (2024), we integrate explicit velocity loss from momentum conservation in Material Point Method (MPM) simulations, ensuring physical realism. We also introduce a material decoder that supports mixed materials through soft gating, enabling the prediction of Mixture-of-Experts Material Constitutive Models (MoEMCMs) and per-region property estimation. The pipeline is optimized end-to-end, coupling perceptual guidance with physical constraints via shared Gaussian features.

The main contributions of our work are as follows:

- We introduce a unified pipeline that generates high-quality 3D objects with physics-grounded motion from text prompts.
- We propose a novel LLM-CoT-Iterative Prompt Refinement method to enhance prompt alignment, along with an innovative multi-view geometry guidance and Densification-by-Adaptive-Splitting mechanism, to generate 3D objects with high-quality appearance and accurate shapes.
- We introduce a novel Mixture-of-Experts Material Constitutive Model prediction to enable mixed constitutive materials, integrating both implicit and explicit physical priors from the video diffusion model and velocity loss to generate physics-grounded motion.
- Experimental results demonstrate that our DiffuPhyGS outperforms other methods in generating realistic, physics-grounded 3D dynamics across a diverse range of materials, with improved visual quality and motion generation.

## 2 RELATED WORK

**Text-to-3D Generation**    As an innovative approach in generative AI, text-to-3D generation enables the generation of 3D models directly from the text prompts. Recent advancements in diffusion models have led to a surge of works utilizing diffusion priors to ensure that generated 3D models align closely with the text prompt descriptions Xu et al. (2023); Hong et al. (2024); Ding et al. (2024); Tang et al. (2023b); Li et al. (2023a); Raj et al. (2023); Chen et al. (2024b). For example, DreamBooth3D Raj et al. (2023) combines Neural Radiance Fields (NeRF) with 2D diffusion priors for efficient optimization. DreamFusion Poole et al. (2022) uses Score Distillation Sampling to align 2D priors with rendered images. Magic3D Lin et al. (2023) employs a coarse-to-fine strategy with diffusion priors and a 3D hash grid to enhance NeRF optimization. Recently, 3D Gaussian Splatting (3DGS) Kerbl et al. (2023) has advanced 3D rendering with fast, detailed point-based representations. Works like GSGEN Chen et al. (2024b) integrate 3DGS with diffusion priors for photorealistic 3D models Chen et al. (2024a); Yi et al. (2024). Despite progress, challenges remain in prompt alignment, multi-view consistency, and visual quality. Our approach addresses these by

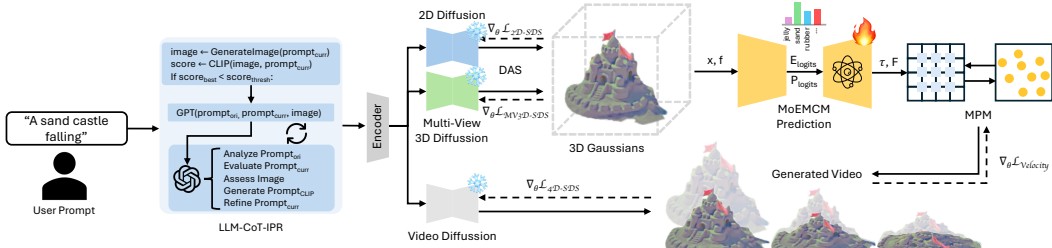

Figure 2: **Pipeline Overview.** *DiffuPhyGS* employs LLM-CoT-IPR for prompt refinement, generates 2D and multi-view 3D diffusion priors to guide GS rendering of 3D Gaussian objects, and enhances quality with DAS. Material properties are derived using a feature encoder and property decoder. MPM deforms Gaussian kernels for physics-based motion rendering, guided by video diffusion model priors and velocity loss.

generating high-quality, prompt-aligned 3D objects with accurate shapes and consistent multi-view appearances.

**Dynamic 3D Generation**   Recent advances in 3D object generation have focused on integrating dynamic behaviors into 3D models Bahmani et al. (2024); Ling et al. (2024); Zhang et al. (2024a); Zeng et al. (2024); Zheng et al. (2024); Cao & Johnson (2023); Shao et al. (2023); Fridovich-Keil et al. (2023); Abou-Chakra et al. (2024). Many approaches leverage video diffusion priors with SDS Bahmani et al. (2024); Zhang et al. (2024a); Zeng et al. (2024) and use dynamic representations like deformable NeRFs Zheng et al. (2024), HexPlane Cao & Johnson (2023), 4D tensor decomposition Shao et al. (2023), K-Planes Fridovich-Keil et al. (2023), or dynamic 3D Gaussians Ling et al. (2024). Multi-view consistency is enhanced through multi-view 4D diffusion models Zhang et al. (2024a); Liang et al. (2024a) or 4D diffusion priors Zeng et al. (2024). However, these methods often lack physics-grounded motion due to insufficient physical priors. Our approach integrates implicit and explicit physical priors to generate physically consistent motion.

**Interactive 3D Dynamics Synthesis**   Recent methods have advanced interactive generation of dynamic 3D objects responding to user inputs (Jiang et al., 2024a; Ling et al., 2024), using 4D Score Distillation Sampling and deformation fields. Physics-aware motion generation under constraints is explored in (Xie et al., 2024; Li et al., 2023b; Jiang et al., 2024b). Notably, PAC-NeRF (Li et al., 2023b), PhysGaussian (Xie et al., 2024), and VR-GS (Jiang et al., 2024b) integrate physics simulations with NeRF and 3D Gaussians for realistic motion but often require manual parameter setting and fixed constitutive models. Recent approaches (Huang et al., 2024; Zhang et al., 2024b; Lin et al., 2025) learn physical properties from video diffusion models, but they require a pre-defined 3D object model. Moreover, OmniPhysGS (Lin et al., 2025) assumes locally homogeneous expert models and relies purely on video priors for motion, PhysDreamer (Zhang et al., 2024b) learns dynamics only in image space from the pretrained video diffusion model, and PhysGaussian (Xie et al., 2024) depends on user-specified physical parameters. In contrast, our DiffuPhyGS provides a unified text-to-3D-to-motion pipeline that directly generates 3D Gaussian assets from text, learns heterogeneous, spatially varying materials via Mixture-of-Experts constitutive models, and explicitly enforces physical constraints during MPM-based simulation, enabling realistic and efficient physics-grounded motion generation.

## 3  METHOD

In this section, we introduce our framework *DiffuPhyGS* (Figure 2) for generating 3D objects with physics-grounded motion. First, *DiffuPhyGS* refines the input prompt with LLM-CoT-IPR and generates the 3D object with guided GS rendering. Next, it learns the object's physical properties using the material property decoder. It then utilizes MPM to deform the Gaussian kernels using both implicit and explicit physical priors, which enables realistic physics-grounded motion. To ensure pipeline cohesion, we employ a shared Gaussian representation with end-to-end joint optimization of material parameters under combined perceptual and physical losses.

## 3.1 PRELIMINARY

3D Gaussian Splatting (3DGS) enables high-quality scene reconstruction with fast training and rendering Kerbl et al. (2023). This point-based method represents scenes using 3D Gaussians, defined by position $x_i$, covariance $\sigma_i$, opacity $\alpha_i$, and spherical harmonic coefficients $c_i$, expressed as $G(x) = e^{-\frac{1}{2}(x)^T \Sigma^{-1}(x)}$. For rendering, 3D Gaussians are projected into 2D space and sorted by depth using tile-based rasterization. Each tile is processed by a thread block, computing pixel colors via alpha-blending:

$$C = \sum_{i \in N} c_i \alpha_i \prod_{j=1}^{i-1} (1 - \alpha_j), \tag{1}$$

where $\alpha_i$ and $c_i$ denote opacity and color of point $i$, and $N$ is the number of Gaussians per tile. Density is managed through pruning and densification Kerbl et al. (2023). We integrate 3DGS into our framework for object representation, extending the GS kernel to include time-dependent $x_i$ and $\sigma_i$ for physics-based motion in generative tasks.

## 3.2 LLM-CoT-ITERATIVE PROMPT REFINEMENT

Text-to-3D generation often produces unsatisfactory results when the input text prompt is overly brief, lengthy, or involves complex logical relationships. This limitation arises primarily from the constrained text comprehension capabilities of the guidance models used in the process. Typically, 3D generation models rely on 2D image generation models, such as diffusion models Rombach et al. (2022); Nichol et al. (2022a), which depend on classifier guidance models like CLIP's text encoder Radford et al. (2021). These classifier guidance models lack advanced natural language understanding and are trained on datasets with simple textual descriptions that lack complex logical information. As a result, the visual concepts they encode are limited, restricting text-to-3D models to perform effectively only with simple prompts.

To address this, we introduce an LLM-based CoT-Iterative Prompt Refinement (LLM-CoT-IPR) module that is used in the text-to-3D stage to optimize the 3D object generation, and we use GPT-4o as the LLM to improve text alignment in the generated 3D object. As outlined in Algorithm 1, we generate an image using *Stable Diffusion*, compute its CLIP score, and, if below the threshold within the maximum iterations, refine the prompt with GPT-4o to maximize the CLIP score.

Using chain-of-thought (CoT) prompting Wei et al. (2022), we instruct GPT-4o to: 1) Analyze the original prompt; 2) Evaluate the current prompt; 3) Assess the generated image; 4) Generate a CLIP prompt; 5) Refine the prompt for optimal length, clarity, and logical complexity. This process mitigates issues like vague prompts (e.g., "a red rose", Figure 5) or overly complex ones (Figure 6), ensuring coherent input for downstream diffusion and GS modules.

## 3.3 TEXT-TO-3D

To generate high-quality 3D objects with realistic shapes and appearances, we adopt 3D Gaussians as our 3D representation, leveraging GS rendering. This is motivated by its point-based nature, ability to produce high-quality rendering results, and fast rendering speed.

**Multi-View 3D Diffusion Prior** Building on previous methods Tang et al. (2023a); Chen et al. (2024b), we employ diffusion priors as rendering guidance. However, unlike these approaches, we introduce a multi-view 3D point cloud diffusion prior to mitigate the Janus problem—where models overfit to specific views, leading to artifacts such as multiple faces or inaccurate geometry. Specifically, we use the 2D diffusion model *MVDream* Shi et al. (2024) to generate multi-view images of the object based on a given prompt. These images are then fed into the image-to-3D-point-cloud diffusion model *Point-E* Nichol et al. (2022b) to create a multi-view 3D point cloud diffusion prior. To use this prior as shape guidance, we apply a 3D Score Distillation Sampling loss Alldieck et al. (2024) to guide the shape optimization process:

$$\mathcal{L}_{\text{shape}} = \mathbb{E}_{\epsilon_I, t} \left[ w_I(t) \left\| \epsilon_\phi(\hat{I}_t; y, t) - \epsilon_I \right\|_2^2 \right] + \mathbb{E}_{\epsilon_X, t} \left[ w_X(t) \left\| \epsilon_\psi(x_t; y, t) - \epsilon_X \right\|_2^2 \right] \cdot \lambda_{\text{MV3D}}, \tag{2}$$

where $x_t$ denotes the noisy Gaussian positions, $\hat{I}$ represents the generated image, $w$ is the weighting function, and $\epsilon$ is the Gaussian noise.

**Densification-by-Adaptive-Splitting** To enhance 3D model visual quality and capture fine details efficiently, we introduce the Densification-by-Adaptive-Splitting (DAS) mechanism in Gaussian Splatting (GS). DAS adaptively refines 3D Gaussians using a 2D appearance diffusion prior from Stable Diffusion Rombach et al. (2022).

DAS computes a per-Gaussian splitting threshold based on local gradient magnitude relative to the global mean and standard deviation. Gaussians in high-gradient regions, indicating fine details, are split if they exceed this threshold, allocating more primitives to detailed areas without overusing resources. The threshold is dynamically adjusted via an adaptation factor to avoid unnecessary splits.

During optimization, the gradient $g_i$ is the L2 norm of the loss gradient with respect to the $i$th Gaussian's mean position, computed via PyTorch automatic differentiation. Gradients are detached to avoid second-order derivative costs and guide splitting in a non-differentiable step post-optimization. The adaptive threshold $\tau_i$ is:

$$\tau_i = \tau_{\text{base}} \left( 1 + \alpha \cdot \frac{g_i - \bar{g}}{\sigma_g + \epsilon} \right), \tag{3}$$

where $\tau_{\text{base}}$ is the baseline threshold, $\alpha$ controls sensitivity, $g_i$, $\bar{g}$, and $\sigma_g$ are the gradient norm, mean, and standard deviation, respectively, and $\epsilon$ prevents division by zero. Gaussians with $g_i \geqslant \tau_i$ are split per the original GS strategy Kerbl et al. (2023), using existing backpropagation gradients efficiently.

The appearance refinement loss is:

$$\mathcal{L}_{\text{appearance}} = \mathbb{E}_{\epsilon_I, t} \left[ w_I(t) \left\| \epsilon_\phi(\hat{I}_t; y, t) - \epsilon_I \right\|_2^2 \right] \cdot \lambda_{\text{SDS}}, \tag{4}$$

where $\hat{I}$ is the generated image and $\lambda_{\text{SDS}}$ is the SDS loss weight. By optimizing with multi-view 3D shape and 2D appearance priors, and applying DAS periodically, our method synthesizes consistent, high-quality 3D models suitable for downstream material encoding.

### 3.4 3D-TO-MOTION

To generate realistic, physics-grounded dynamics of 3D Gaussians, we employ the Material Point Method (MPM) through 3DGS rendering, guided by both implicit and explicit physical priors: a video diffusion prior and a velocity loss. We use a material feature encoder to extract features from the 3D Gaussians and a material property decoder with soft gating to predict the material properties of the Gaussians using Mixture-of-Experts Material Constitutive Models (MoEMCMs). The objective is to optimize the learnable material property parameters $\theta$.

**Material Point Method** We utilize MPM Sulsky (1994) to simulate the material behaviors of objects under various physical forces and deformations. We employ the 3D Gaussians to represent the discrete particles, and we use a deformation map $\phi(X, t)$ to describe the motion of a particle's position $x_i$ at the time $t$ Xie et al. (2024). Local transformations are defined using the gradient of the deformation map as $F(X, t) = \nabla_X \phi(X, t)$, which decomposes into elastic and plastic components: $F = F^E F^P$. To align with continuum mechanics, updates to the deformation map $\phi$ conform to the conservation of mass and momentum Sulsky (1994):

$$\frac{D\rho}{Dt} + \rho \nabla \cdot v = 0, \quad \rho \frac{Dv}{Dt} = \nabla \cdot \tau + f^{\text{ext}}, \tag{5}$$

where

$$\tau = \frac{1}{\det(F)} \frac{\partial \Psi}{\partial F}(F^E)(F^E)^T \tag{6}$$

represents the Cauchy stress tensor. $\Psi(\cdot)$ is the hyperelastic energy density function, determined by the material-specific elasticity model. Depending on the material-specific plasticity model, $F^E = M(F^E)$, where $M(\cdot)$ is the return mapping that enforces plasticity constraints on $F^E$.

The material properties are only determined by the $\Psi(\cdot)$, $M(\cdot)$, and the physical parameters $\gamma$. Unlike previous works Xie et al. (2024); Li et al. (2023b); Jiang et al. (2024b) that fix these material properties, we treat them as learnable material models for the Gaussians to estimate the material property parameters: $\Psi_{\theta_{\text{el}}}$, $M_{\theta_{\text{pl}}}$, $\gamma_{\theta_{\text{phy}}}$, where $\theta_{\text{el}}$, $\theta_{\text{pl}}$, and $\theta_{\text{phy}}$ are the elasticity, plasticity, and physical parameters, respectively. This learnable parameterization enables joint optimization with upstream Gaussian features and downstream dynamics.

**Mixture-of-Experts Material Constitutive Models**  In MPM physics simulations, local interactions between neighboring Gaussians govern material behaviors such as elasticity (how a material deforms under stress and returns to its original shape when stress is removed) and plasticity (permanent deformation under stress). To accurately capture these material behaviors, we collect a set of material constitutive models Xie et al. (2024); Zong et al. (2023) (Appendix A), covering a wide range of material types to simulate physics-grounded motion. Unlike previous methods Lin et al. (2025) that assume homogeneous material properties within local neighborhoods, we adopt heterogeneous material properties to capture mixed materials within local neighborhoods of the object. Specifically, our MoEMCMs enable per-Gaussian material customization by blending multiple constitutive models with learned weights, allowing for spatially varying material compositions (e.g., a single object with regions of rubber-like elasticity and jelly-like plasticity). This facilitates realistic simulations of objects with mixed materials, such as a soft toy with a rigid core and flexible exterior.

**Material Feature Encoder**  To predict per-Gaussian elasticity and plasticity properties, we design a material feature encoder inspired by the architecture of Point-BERT Yu et al. (2022), which is effective for processing point cloud data by capturing local geometric structures and contextual relationships through grouping and self-attention mechanisms. We encode local neighborhoods of 3D Gaussian features, allowing the model to learn material-specific representations from scratch during training. This choice is motivated by the need to infer spatially varying material properties from the 3D Gaussian features $\{X_i, \Sigma_i, \alpha_i, c_i\}$, which implicitly encode object structure and texture cues that correlate with physical behaviors.

The encoder partitions the 3D Gaussian features into local neighborhood features using farthest point sampling (FPS) for group centers and K-Nearest Neighbors (KNN) for grouping. These grouped features are then processed through convolutional layers for initial encoding, followed by transformer blocks to aggregate contextual information. The encoded features are mapped to logits as:

$$Encoder(x, f) \rightarrow (e_{\text{logits}}, p_{\text{logits}}), \tag{7}$$

where $x$ is the Gaussian positions, $f$ is the concatenated features, and $e_{\text{logits}}$, $p_{\text{logits}}$ influence elasticity and plasticity behaviors. The encoder is trained end-to-end with the full pipeline, using gradients from both SDS and velocity losses to optimize material predictions jointly with dynamics.

**Material Property Decoder**  We assume heterogeneous material properties in local object neighborhoods and decode contribution logit scores by assigning Mixture of Experts Material Constitutive Models (MoEMCMs) to Gaussians using softmax-derived weights:

$$Decoder(F, \text{logits}) = \sum_{c=1}^{C} P_c(F_n) \cdot w_{n,c}, \tag{8}$$

where

$$w_{n,c} = \frac{\exp(\text{logits}_{n,c})}{\sum_{c'=1}^{C} \exp(\text{logits}_{n,c'})}, \tag{9}$$

with $w_{n,c}$ as the weight for material model category $c$ at Gaussian $n$, logits representing contribution scores ($e_{\text{logits}}$ or $p_{\text{logits}}$), $P_c$ the material constitutive model, and $F_n$ the deformation gradient. Material models $\Psi(\cdot)$ and $M(\cdot)$ are decoded as:

$$\begin{aligned} \tau &= \Psi(F) = Decoder(F, e_{\text{logits}}), \\ F &= M(F) = Decoder(F, p_{\text{logits}}). \end{aligned} \tag{10}$$

This enables heterogeneous material compositions, with distinct properties across regions (e.g., stiff base, compliant top). Following Xie et al. (2024); Zong et al. (2023), we use 7 material constitutive models: 3 hyperelastic density functions and 4 plasticity return mappings (please see supplementary material for details). MoEMCMs capture diverse material behaviors for realistic physics-grounded motion.

**Video Diffusion SDS Guidance**   To ensure realistic physics-based motion generation, we adopt an implicit physical prior from video diffusion SDS to guide the 3DGS rendering. Given generated positions $\hat{x}$ and deformation gradients $\hat{F}$, we leverage ModelScope Wang et al. (2023) to obtain the video diffusion prior, guiding the generation of motion clip $\hat{V}$ with learnable material parameters $\theta$:

$$\nabla_\theta \mathcal{L}_{\text{SDS}} = \mathbb{E}_{\xi,\epsilon} \left[ \omega(\xi) \left( \hat{\epsilon}_\Phi(\hat{V}; \xi, y) - \epsilon \right) \frac{\partial \hat{V}}{\partial \hat{x}, \hat{F}} \frac{\partial \hat{x}, \hat{F}}{\partial \theta} \right]. \tag{11}$$

**Velocity Loss**   To enhance physical realism in motion, we incorporate a velocity loss based on Newton's Second Law, $F = m \cdot \frac{dv}{dt}$, discretized in the Material Point Method (MPM) as:

$$\Delta v = \frac{F \cdot \Delta t}{m}, \tag{12}$$

where $F$ is the applied force, $m$ is the mass, and $\Delta t$ is the time step.

The velocity loss enforces momentum conservation by penalizing deviations between the actual velocity change ($\Delta v_{\text{actual},i} = v_{i,t+1} - v_{i,t}$) from the MPM solver and the expected velocity change:

$$\Delta v_{\text{expected},i} = \frac{(F_{\text{stress},i} + F_{\text{ext},i})\Delta t}{m_i}, \tag{13}$$

where $F_{\text{stress},i} = -\Delta t \, V_i \sum_j \sigma_i : \nabla w_{ij}$ represents internal stress forces, and $F_{\text{ext},i} = m_i \cdot g$ denotes external gravity forces. The velocity loss is defined as the mean squared error:

$$\mathcal{L}_{\text{vel}} = \frac{1}{N} \sum_{i=1}^{N} \|\Delta v_{\text{actual},i} - \Delta v_{\text{expected},i}\|^2. \tag{14}$$

The total loss is obtained by adding the velocity loss to the training objective, weighted by $\lambda_{\text{vel}}$:

$$\mathcal{L}_{\text{total}} = \mathcal{L}_{\text{SDS}} + \lambda_{\text{vel}} \cdot \mathcal{L}_{\text{vel}}. \tag{15}$$

## 4 EXPERIMENTS

### 4.1 IMPLEMENTATION DETAILS

**Setup**   We implement the 3D Gaussian Splatting using PyTorch, following the optimization pipeline from Kerbl et al. (2023). To generate physics-grounded motion, we build upon MPM Sulsky (1994); Xie et al. (2024). All experiments are conducted on an NVIDIA RTX 3090 GPU. We set the loss weights $\lambda_{\text{MV3D}} = 0.1$, $\lambda_{\text{DAS}} = 0.1$, and $\lambda_{\text{vel}} = 0.2$.

**Metrics**   For dynamic scenes without real-world ground truth, we follow standard practice in the domain Lin et al. (2025) and use a randomly initialized, frozen model as a *reference* baseline rather than a physical ground-truth video. For each prompt and method, we generate a video with the trained model and a reference video from the same input using the reference model. We use CLIP-SIM Radford et al. (2021) to measure prompt consistency, calculated as the average cosine similarity between text prompt and video frame embeddings. We adopt $\text{Diff}_{SSIM}$ and $\text{Diff}_{CLIP}$ Lin et al. (2025) to assess expressiveness and robustness by comparing videos from trained and randomly initialized models. Higher CLIPSIM, $\text{Diff}_{SSIM}$, and $\text{Diff}_{CLIP}$ indicate better performance. Fréchet Video Distance (FVD) Unterthiner et al. (2018) evaluates video quality, and LAION-Aesthetic scores Schuhmann et al. (2022) assess the aesthetic quality of generated 3D objects.

**Baselines**   We compare our method with three baseline methods: 1) **PhysDreamer** Zhang et al. (2024b), which learns dynamics priors from videos generated by diffusion models to simulate object dynamics; 2) **OmniPhysGS** Lin et al. (2025), which assigns homogeneous expert constitutive models to the object's local neighborhoods and uses physical priors learned from video diffusion models to generate object motion; 3) **PhysGaussian** Xie et al. (2024), which generates object motion based on user-defined parameters of physical dynamics. Since these methods do not support generating 3D objects directly from prompts, we focus on evaluating their supported motion generation using 3D models produced by our method. We do not directly compare with generic 2D video

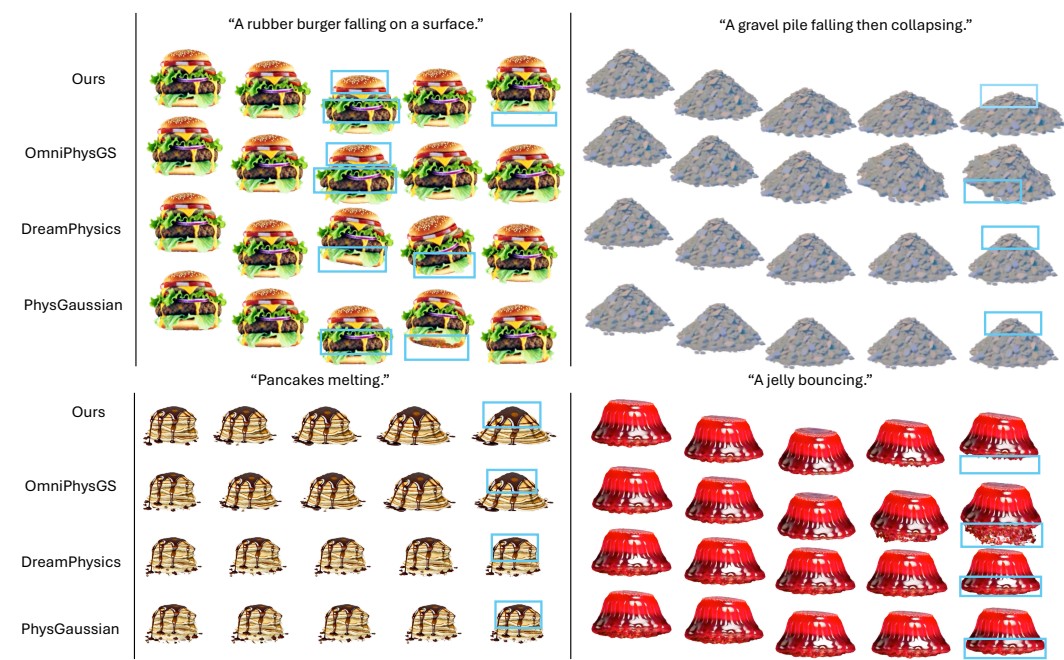

Figure 3: Qualitative comparisons of our physics-grounded motion results with other baselines. Compared with baseline methods, our DiffuPhyGS generates more realistic and physically consistent behaviors (e.g., elastic bouncing, granular collapse, viscous melting), whereas baselines produce results with weaker or unrealistic motions (e.g., highlighted rectangular areas) that deviate from expected physical responses.

Table 1: User study results. The best results are in **bold**.

| Criteria/Method | Ours | OmniPhysGS | DreamPhysics | PhysGaussian |
|---|---|---|---|---|
| Motion Quality | **2.55 ± 0.25** | 2.00 ± 0.34 | 1.91 ± 0.84 | 1.79 ± 0.71 |
| Visual Quality | **2.46 ± 0.12** | 2.06 ± 0.28 | 2.00 ± 0.39 | 1.94 ± 0.27 |
| Average | **2.51 ± 0.19** | 2.03 ± 0.31 | 1.96 ± 0.62 | 1.86 ± 0.49 |

generation methods (e.g, diffusion-based models Wu et al. (2025); Wan et al. (2025)), since they operate purely in 2D without explicit 3D geometry, physical priors, and cannot be subjected to the same forces and conditions as our method. Instead, we focus on 3D object-based methods that share the same 3D representations, enabling fair assessment of physical plausibility under comparable conditions.

## 4.2 QUALITATIVE EVALUATION

We demonstrate the qualitative performance of our framework by generating 3D objects with diverse and diagnostically challenging physics-based dynamics (e.g., **rubber**, **jelly**, **sand**, **granular**, **melting**, **fracture**). These test cases are deliberately chosen because they are particularly revealing of physical plausibility, and they form a shared subset of motions that all compared methods can reasonably handle, ensuring a fair and controlled evaluation.

**Qualitative Comparisons** Figure 3 highlights our framework's ability to synthesize 3D objects with realistic shapes, appearances, and physics-grounded motion based on given prompts. For fair comparison, we follow the evaluation protocol adopted in prior works and focus on canonical motion prompts that baselines such as OmniPhysGS, DreamPhysics, and PhysGaussian are designed to address, while also being highly sensitive to violations of basic physics. Compared to our framework, OmniPhysGS's falling rubber burger lacks elasticity, barely bouncing on impact, and its gravel pile bounces unrealistically instead of collapsing. DreamPhysics fails to capture rubber friction, causing

slipping, and its jelly lacks bouncing motion. PhysGaussian's rubber burger appears rigid without deformation, and its pancakes show no melting motion. Beyond these benchmark prompts, our framework, enabled by the proposed MoEMCMs and heterogeneous material fields, can generate richer behaviors, which we illustrate in Appendix 4 (Figure 4) and in the supplementary videos.

**User Study** We conducted a user study with 20 participants to evaluate the human-perceived quality of the generated 3D object motion videos. For each of the 4 methods, we generated 5 video clips, resulting in a total of 20 videos. Participants rated each video using the Mean Opinion Score (MOS), with scores ranging from 1 (Bad) to 5 (Excellent), based on two criteria: 1) motion quality, and 2) visual quality.

As reported in Table 1, our framework consistently achieves the highest MOS across all criteria: it outperforms the strongest baseline by a clear margin in both motion quality and visual quality leading to the best average score overall. These user study results not only demonstrate a considerably better perceived quality compared to prior methods, but also align well with the quantitative evaluation results presented in the paper.

### 4.3 QUANTITATIVE EVALUATION

**Quantitative Comparisons** Table 2 shows the quantitative evaluation results for prompts such as *"A rubber burger falling on a surface"*, *"A gravel pile falling then collapsing"*, *"Pancakes melting"*, and *"A jelly bouncing"*. Across these diverse materials, our method obtains the best average scores on CLIPSIM, $\text{Diff}_{SSIM}$, and $\text{Diff}_{CLIP}$, indicating stronger alignment with the text prompt, more expressive and faithful physical behavior, and improved robustness compared to all baselines. Although OmniPhysGS achieves the lowest average FVD (a $2.5\%$ improvement over ours), it consistently underperforms our method on the physics- and content-aware metrics, trailing by $0.2\%$ in CLIPSIM, $45.1\%$ in $\text{Diff}_{SSIM}$, and $10.6\%$ in $\text{Diff}_{CLIP}$. This demonstrates that our approach achieves a better overall trade-off between perceptual video quality and accurate, controllable physical dynamics.

**Efficiency and Memory Usage** Table 3 reports the total time, average epoch time, and average peak GPU memory for the 3D-to-motion stage. PhysGaussian (PhysGS) attains the lowest total time and peak memory usage, which is expected since it does not learn material or dynamics parameters and thus avoids the overhead of optimization. Among the learning-based methods (DreamPhysics, OmniPhysGS, and DiffuPhyGS), our DiffuPhyGS achieves the lowest total time and average peak memory, and the second-best average epoch time (slightly slower than DreamPhysics). Overall, DiffuPhyGS offers a favorable trade-off between efficiency and capability, because it maintains competitive computational cost while providing a full text-to-3D-to-motion pipeline with physics-grounding.

### 4.4 ABLATION STUDY

**LLM-CoT-Iterative Prompt Refinement** Ablation studies show that removing LLM-CoT-IPR lowers CLIPSIM and LAION scores (Table 6), highlighting its role in prompt consistency and aesthetic quality (Figure 11).

**Multi-View 3D Diffusion Prior** Incorporating the multi-view 3D diffusion prior enhances geometric accuracy in 3D objects, as shown in Figure 9 and Figure 7, improving shape quality.

**Densification-by-Adaptive-Splitting** We conduct ablation studies to evaluate the necessity of DAS. Figure 8 shows that with DAS, the number of Gaussians increases steadily from the $6,000$th to the $10,000$th step, with consistent increments every $1,000$ steps. Without DAS, the number of Gaussians increases rapidly with larger increments, indicating over-allocation of Gaussians. DAS adaptively splits Gaussians in regions with higher gradients, ensuring efficient allocation and stable growth of Gaussians, which is crucial for rendering finer detail (Figure 10).

**Physics-Grounded Motion Generation** As shown in Table 4, removing any of the proposed components consistently degrades performance across all metrics, underscoring their importance for

Table 2: Results of quantitative evaluation. "↑": higher is better. "↓": lower is better. The best results are in **bold**. Each cell shows the percentage difference relative to our method DiffuPhyGS.

| Method | Metric | Burger | Gravel | Melting | Jelly | Average |
|---|---|---|---|---|---|---|
| PhysGaussian Xie et al. (2024) | CLIPSIM↑ | 0.2888 -5.3% | **0.2966** +1.6% | 0.2496 -1.0% | 0.3035 +0.3% | 0.2846 -1.2% |
| | Diff$_{SSIM}$↑ | 0.0757 -42.6% | 0.0786 +10.5% | 0.0814 +16.5% | 0.0807 +33.6% | 0.0791 -5.0% |
| | Diff$_{CLIP}$↑ | 1.3375 +33.5% | 1.7621 -18.5% | 0.9889 -28.6% | 1.2418 -3.6% | 1.3326 -8.7% |
| | FVD↓ | 30.9738 -16.7% | 27.0694 -8.3% | **22.1761** +25.7% | 22.3910 -78.3% | 25.6526 -9.2% |
| DreamPhysics Huang et al. (2024) | CLIPSIM↑ | 0.2942 -3.5% | 0.2913 -0.2% | 0.2493 -1.1% | **0.3059** +1.1% | 0.2852 -1.0% |
| | Diff$_{SSIM}$↑ | 0.0772 -41.4% | **0.0789** +11.0% | **0.0832** +19.0% | **0.0810** +34.1% | 0.0801 -3.8% |
| | Diff$_{CLIP}$↑ | **1.3624** +36.0% | 1.7302 -20.0% | 0.9874 -28.7% | 1.2517 -2.8% | 1.3329 -8.7% |
| | FVD↓ | 28.1310 -6.0% | 26.8209 -7.3% | 22.2417 +25.5% | 22.2282 -77.0% | 24.8554 -5.8% |
| OmniPhysGS Lin et al. (2025) | CLIPSIM↑ | 0.2986 -2.1% | 0.2932 +0.4% | **0.2559** +1.5% | 0.3013 -0.5% | 0.2872 -0.2% |
| | Diff$_{SSIM}$↑ | 0.0600 -54.5% | 0.0553 -22.2% | 0.0659 -5.7% | 0.0014 -97.7% | 0.0457 -45.1% |
| | Diff$_{CLIP}$↑ | 1.2643 +26.2% | 1.7727 -18.0% | 1.1815 -14.6% | 0.9984 -22.5% | 1.3042 -10.6% |
| | FVD↓ | **14.1274** +46.8% | 29.8549 -19.4% | 24.2862 +18.6% | 23.3247 -85.7% | **22.8981** +2.5% |
| **DiffuPhyGS (Ours)** | CLIPSIM↑ | **0.3049** | 0.2920 | 0.2520 | 0.3027 | **0.2879** |
| | Diff$_{SSIM}$↑ | **0.1318** | 0.0711 | 0.0699 | 0.0604 | **0.0833** |
| | Diff$_{CLIP}$↑ | 1.0016 | **2.1627** | 1.3843 | 1.2883 | 1.4592 |
| | FVD↓ | 26.5382 | **24.9928** | 29.8428 | **12.5603** | 23.4835 |

Table 3: Results of efficiency and memory usage evaluation. The best results are in **bold**, the second best are underlined.

| Method/Metric | Total Time (s)↓ | Avg. Epoch Time (s)↓ | Avg. Peak Mem. (MB)↓ |
|---|---|---|---|
| DreamPhysics | 12.5 | 7.7 | 9043.2 |
| OmniPhyGS | 11.8 | 8.3 | 11742.1 |
| PhysGS | **4.5** | **0.04** | **439.7** |
| DiffuPhyGS (Ours) | 10.3 | 8.2 | 8597.2 |

Table 4: Ablation study results for physics-grounded motion generation. The best mean results are in **bold**. All metrics are reported as mean $\pm$ standard deviation.

| Setting/Metric | CLIPSIM↑ | Diff$_{SSIM}$↑ | Diff$_{CLIP}$↑ | FVD↓ |
|---|---|---|---|---|
| DiffuPhyGS [Full] | **0.2879** $\pm$ 0.0031 | **0.0833** $\pm$ 0.0045 | **1.4592** $\pm$ 0.052 | **23.4835** $\pm$ 1.3 |
| w/o MoEMCMs | 0.2613 $\pm$ 0.0036 | 0.0460 $\pm$ 0.0033 | 0.9984 $\pm$ 0.047 | 31.2345 $\pm$ 1.6 |
| w/o Velocity Loss | 0.2622 $\pm$ 0.0034 | 0.0460 $\pm$ 0.0031 | 1.0016 $\pm$ 0.049 | 27.8921 $\pm$ 1.4 |
| w/o Video Diffusion Prior | 0.2622 $\pm$ 0.0038 | 0.0460 $\pm$ 0.0030 | 1.0016 $\pm$ 0.051 | 29.4567 $\pm$ 1.5 |
| w/o LLM-CoT-IPR | 0.2683 $\pm$ 0.0042 | 0.01397 $\pm$ 0.0021 | 0.9865 $\pm$ 0.054 | 124.4351 $\pm$ 2.3 |

video quality, expressiveness, and robustness (Figure 12). In particular, excluding the MoEMCMs or LLM-CoT-IPR leads to pronounced drops in CLIPSIM, Diff$_{SSIM}$, and Diff$_{CLIP}$, while ablating the velocity loss or the video diffusion prior yields smaller but systematic differences in these prompt-consistency metrics and noticeably worse FVD compared to the full model. The reported mean $\pm$ standard deviation over three random seeds further indicates that these performance gaps are stable rather than due to stochastic variation in the SDS optimization.

## 5 CONCLUSION

In this paper, we present an innovative pipeline that generates high-quality 3D objects with realistic, physics-aware motion based on text prompts. Our pipeline integrates natural language processing, generative modeling, and physics simulation, pushing the boundaries of 3D dynamic generation. This advancement paves the way for transformative applications across diverse industries, including filmmaking, virtual and augmented reality, gaming, and beyond. Future work could explore the incorporation of relighting techniques and a wider variety of material types to further enhance the realism and versatility of the generated motion.

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

## A APPENDIX

### A.1 MATERIAL POINT METHOD

The Material Point Method (MPM) is a computational physics technique used to simulate material behaviors under various physical forces and deformations Sulsky (1994). In MPM, a material body is discretized into a collection of Lagrangian particles. Each particle carries a set of quantities, including position $x_i^n$, mass $m_i$, velocity $v_i^n$, Kirchhoff stress tensor $K_i^n$, deformation gradient $F_i^n$, and affine momentum $A_i^n$ on particle $i$ at time $t^n$. At time $t^n$, let $x_j^n$, $m_j$, and $v_j^n$ represent the position, mass, and velocity on grid node $j$, which facilitate the computation of deformations

**"A rubber corgi falling to the ground."**

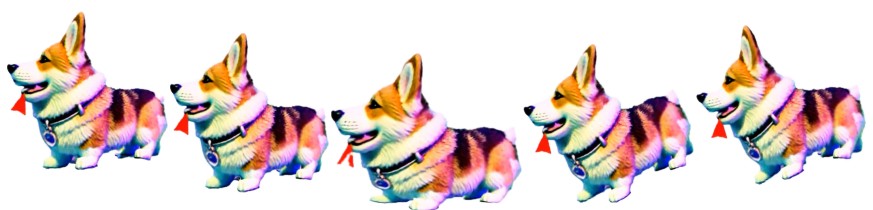

**"A sponge tearing apart."**

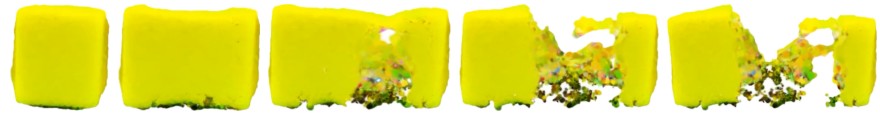

**"A pineapple made of jelly bouncing on the surface."**

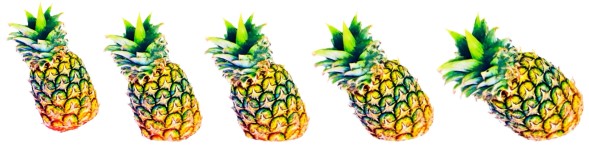

**"A blue and white porcelain vase made of sand collapsing."**

**"A honey jar melting."**

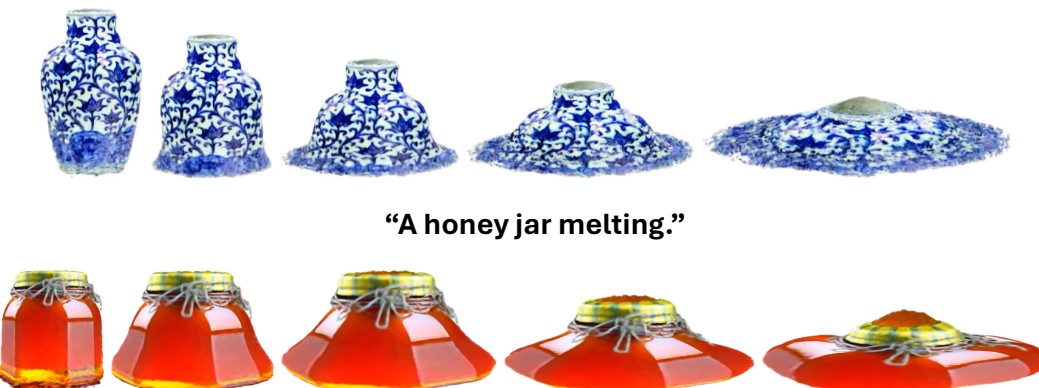

Figure 4: Additional qualitative results.

and applied forces on the material body. Due to the conservation of mass, particle mass remains invariant. At each time step, MPM performs a two-way transfer process: 1) Particle-to-Grid and 2) Grid-to-Particle.

**Particle-to-Grid Transfer** During this process, the mass and momentum of particles are transferred to the grid nodes Xie et al. (2024). The mass $m_j^n$ at a grid node $j$ is calculated as:

$$m_j^n = \sum_i w_{ji}^n m_i, \tag{16}$$

where $w_{ji}^n$ is the interpolation weight derived from a B-spline kernel. Using the APIC momentum transfer method Jiang et al. (2015), the momentum at grid node $j$ is updated as:

$$m_j^n v_j^n = \sum_i w_{ji}^n m_i (v_i^n + A_i^n (x_j - x_i^n)). \tag{17}$$

Table 5: Physical Parameters.

| Notation | Meaning / Definition | |
|---|---|---|
| $E$ | Young's modulus ($E$) | |
| $\nu$ | Poisson's Ratio ($\nu$) | |
| $\mu$ | Shear modulus | $\frac{E}{2(1+\nu)}$ |
| $\lambda$ | Lamé modulus | $\frac{E\nu}{(1+\nu)(1-2\nu)}$ |

Based on the internal and external forces acting on the particles, the grid velocity $v_j^{n+1}$ at the next time step is updated as:

$$v_j^{n+1} = v_j^n - \frac{\Delta t}{m_j} \sum_i K_i^n \nabla w_{ji}^n V_i^0 + \Delta t g, \tag{18}$$

where $g$ is the gravitational acceleration.

**Grid-to-Particle Transfer** In this stage, the updated velocities and momentum from the grid nodes are transferred back to the particles Xie et al. (2024); Jiang et al. (2016). The velocity $v_i^{n+1}$, position $x_i^{n+1}$, affine momentum $A_i^{n+1}$, and deformation gradient $F_i^{n+1}$ of particle $i$ at the new time step are updated as follows:

$$
\begin{aligned}
v_i^{n+1} &= \sum_j v_j^{n+1} w_{ji}^n, \\
x_i^{n+1} &= x_i^n + \Delta t \, v_i^{n+1}, \\
A_i^{n+1} &= \frac{12}{\Delta x^2(b+1)} \sum_j w_{ji}^n v_j^{n+1} \left(x_j^n - x_i^n\right)^T, \\
\nabla v_i^{n+1} &= \sum_j v_j^{n+1} \left(\nabla w_{ji}^n\right)^T, \\
F_i^{n+1} &= M\!\left(\left(I + \nabla v_i^{n+1}\right) F_i^n\right), \\
K_i^{n+1} &= K(F_i^{n+1}).
\end{aligned}
\tag{19}
$$

Here, $b$ denotes the B-spline degree, and $\Delta x$ represents the Eulerian grid spacing. The calculation of the deformation adjustment mapping $M(\cdot)$ and the Kirchhoff stress tensor $K$ are detailed in the next section.

### A.1.1 MATERIAL CONSTITUTIVE MODELS

We compile a set of material constitutive models from previous work Xie et al. (2024); Zong et al. (2023); Lin et al. (2025), which describe various material behaviors, including those exhibiting elasticity or plasticity. The essential physical parameters for these materials are listed in Table 5.

### A.1.2 ELASTICITY MODELS

We employ the Kirchhoff stress tensor $K = \frac{\partial \Psi}{\partial F}$ to map $F$ to $K$, in order to express the stress-strain relationship.

**Fixed Corotated Elasticity** Following previous work Stomakhin et al. (2012), we define fixed corotated elasticity as:

$$K = 2\mu(F - R) + \lambda J(J - 1)F^{-T}, \tag{20}$$

where $R$ is obtained from the polar decomposition of $F = RS$, and $J$ is the determinant of $F$.

**Neo-Hookean Elasticity** We define Neo-Hookean elasticity, following Bonet & Wood (1997), as:

$$K = \mu(F - F^{-T}) + \lambda \log(J)F^{-T}. \tag{21}$$

**StVK Elasticity** We define the StVK elasticity, based on Barbič & James (2005), as:

$$K = U(2\mu\Sigma^{-1}ln\Sigma + \lambda tr(ln\Sigma)\Sigma^{-1})V^T, \tag{22}$$

where $U$, $\Sigma$, and $V$ are derived from the singular value decomposition of $F = U\Sigma V^T$.

### A.1.3 PLASTICITY MODELS

We employ the return mapping function $M(\cdot)$ to transform the current deformation gradient to the final deformation gradient $F$.

**Identity Plasticity** Most purely elastic materials employ the identity plasticity as:

$$M(F) = F. \tag{23}$$

**Drucker-Prager Plasticity** Following Drucker & Prager (1952); Chen et al. (2021), we define Drucker-Prager plasticity as:

$$M(F) = UZ(\Sigma)V^T, \tag{24}$$

$$Z(\Sigma) = \begin{cases} 1 & \text{sum}(\epsilon) > 0 \\ \Sigma & \delta_\gamma \leqslant 0 \text{ and sum}(\epsilon) \leqslant 0 \\ \exp(\epsilon - \delta\gamma\frac{\hat{\epsilon}}{\|\hat{\epsilon}\|}) & \text{otherwise}, \end{cases} \tag{25}$$

where $\epsilon = \log(\Sigma)$.

**von Mises Plasticity** We define von Mises plasticity, following Mises (1913); Huang et al. (2021), as:

$$M(F) = UZ(\Sigma)V^T, \tag{26}$$

$$Z(\Sigma) = \begin{cases} \Sigma, & \delta\gamma \leqslant 0, \\ \exp\left(\epsilon - \delta\gamma\frac{\hat{\epsilon}}{\|\hat{\epsilon}\|}\right), & \text{otherwise}. \end{cases} \tag{27}$$

**Fluid Plasticity** We define fluid plasticity, following Stomakhin et al. (2014); Gao et al. (2018), as:

$$M(F) = J^{1/3}I. \tag{28}$$

### A.2 METRICS

We adopt the `ViT-B/32` model of CLIP Radford et al. (2021) to calculate the CLIPSIM Wu et al. (2021) score as:

$$\text{CLIPSIM} = \frac{1}{N}\sum_{n=1}^{N} CLIP(\hat{I}_n, y), \tag{29}$$

where $\hat{I}_n$ is the $n$-th frame of the generated video and $y$ is the text prompt. A higher CLIPSIM indicates better alignment between the video and the text.

Following Lin et al. (2025), we define $\text{Diff}_{SSIM}$ and $\text{Diff}_{CLIP}$ as:

$$\text{Diff}_{SSIM} = 1 - \frac{1}{N}\sum_{n=1}^{N} \text{SSIM}(I'_n, \hat{I}_n), \quad \text{Diff}_{CLIP} = \frac{\text{CLIPSIM}}{\text{CLIPSIM}'}, \tag{30}$$

where $I'_n$ is the $n$-th frame of the video generated by a randomly initialized model, $SSIM$ is the structural similarity index Wang et al. (2004), and CLIPSIM$'$ is the CLIPSIM of the randomly initialized model. Higher values of $\text{Diff}_{SSIM}$ and $\text{Diff}_{CLIP}$ indicate greater expressiveness and robustness of the model.

Table 6: Ablation study results of LLM-CoT-IPR. The best results are in **bold**.

| Setting/Metric | CLIPSIM↑ | LAION↑ |
|---|---|---|
| DiffuPhyGS [Full] | **0.2879** | **33.7781** |
| w/o LLM-CoT-IPR | 0.2683 | 17.1845 |

## A.3 SCORE DISTILLATION SAMPLING

Score Distillation Sampling (SDS), introduced in DreamFusion Poole et al. (2022), is a technique that leverages a 2D diffusion prior to optimize an image generator based on probability density distillation. The image generator, parameterized by parameters $\theta$, is represented as $g(\theta)$. To optimize $\theta$ such that the generated image $x = g(\theta)$ resembles a sample from the pre-trained, frozen 2D diffusion model, the SDS loss gradient for optimizing $\theta$ is formulated as:

$$\nabla_\theta \mathcal{L}_{\text{SDS}}(\phi, x = g(\theta)) \triangleq \mathbb{E}_{t,\epsilon} \left[ w(t) \left( \hat{\epsilon}_\phi(z_t; y, t) - \epsilon \right) \frac{\partial x}{\partial \theta} \right], \tag{31}$$

where $\hat{\epsilon}_\phi(z_t; y, t)$ is the noise predicted by the pre-trained 2D diffusion model with text prompt $y$ at time step $t$, $\epsilon$ is the true noise at the time step, $\frac{\partial x}{\partial \theta}$ is the derivative of the generated image with respect to the generator's parameters $\theta$, and $w(t)$ is a weighting function from DDPM Ho et al. (2020). This loss function aligns the scores (or gradients) of the image generator and the 2D diffusion model by optimizing the loss with respect to $\theta$, enabling efficient use of the 2D diffusion prior to guide 3D model generation.

## A.4 LLM-CoT-IPR ALGORITHM AND PROMPTS

We provide LLM-CoT-IPR pseudo-code 1 and example prompts, including an overly brief prompt illustrated in Figure 5 and a complex prompt shown in Figure 6.

---

**Algorithm 1** LLM-CoT-Iterative Prompt Refinement

---

1: **function** LLM-CoT-IPR(ori_prompt, score_thresh, max_iter)
2:     $curr\_prompt \leftarrow ori\_prompt$
3:     $best\_prompt \leftarrow ori\_prompt$
4:     $best\_score \leftarrow -\infty$
5:     **for** each iteration from 1 to $max\_iter$ **do**
6:         $image \leftarrow GenerateImage(curr\_prompt)$
7:         $score \leftarrow CLIP(image, curr\_prompt)$
8:         **if** $score > best\_score$ **then**
9:             $best\_score \leftarrow score$
10:            $best\_prompt \leftarrow curr\_prompt$
11:         **end if**
12:         **if** $best\_score \geqslant score\_thresh$ **then**
13:            Return $best\_prompt$
14:         **end if**
15:         $curr\_prompt \leftarrow GPT(ori\_prompt, curr\_prompt, image)$
16:     **end for**
17:     Return $best\_prompt$
18: **end function**

---

System Prompt

Given the original_prompt, current_prompt, and an image generated from current_prompt, your goal is to refine the current_prompt to better align with the original_prompt while improving clarity and detail. Follow these steps to refine the prompt:

    1. Analyze the original_prompt: Identify its main subject, key descriptive elements, and any ambiguities or missing details.
    2. Evaluate the current_prompt: Compare it to the original_prompt. Determine if it's too short, too long, or misaligned in any way. Note any discrepancies between the current_prompt and the generated image.
    3. Assess the generated image: Describe the visual features of the image (e.g., colors, shapes, textures) and how well they match the current_prompt and original_prompt.
    4. Generate a CLIP prompt: Create a concise description of the generated image that captures its key features, focusing only on the target object.
    5. Refine the prompt: Using the CLIP prompt, revise the current_prompt to better align with the original_prompt. Ensure the refined_prompt follows these rules:
        – Describe only the target object (exclude background, add 'no background').
        – Provide at least 7 values, each with at least 3 words, separated by commas.
        – Add more detail if the current_prompt is too short; simplify if it's too complex, while keeping the main information from the original_prompt.
        – Reduce ambiguity and logical complexity.

    After completing these steps, present your chain of thought reasoning, then output "refined_prompt: object, value1, value2, value3, value4, value5, value6, value7, no background"

Q: 1. original_prompt: "A red rose"; 2. current_prompt: "A red rose"; 3. the generated image (shows a simple red rose with no intricate details).

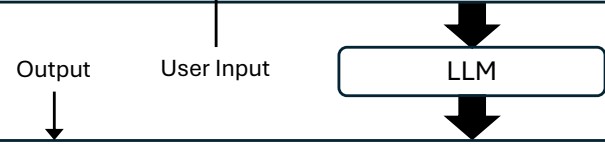

Output    User Input    LLM

A: chain of thought:
    1. Analyze the original_prompt: The main subject is a red rose. It's simple and focuses on color (red) and type (rose), with no additional details.
    2. Evaluate the current_prompt: "A red rose" matches the original_prompt exactly, but it's very short and lacks detail. The generated image likely shows a basic red rose, which may not fully capture potential user expectations for more specifics.
    3. Assess the generated image: The image shows a red rose with smooth petals, uniform red color, and no background or intricate features.
    4. Generate a CLIP prompt: "Red rose with smooth petals, uniform red color, simple shape, no background."
    5. Refine the prompt: The current_prompt is already clear but too brief. I'll add more detail to match potential user expectations while keeping it simple. The refined prompt should describe the rose's appearance more vividly.

refined_prompt: Rose, bright red petals, smooth petal surface, rounded bloom shape, fresh green stem, subtle thorn details, natural floral scent, no background

Figure 5: LLM-CoT-IPR overly brief prompt.

System Prompt

Given the original_prompt, current_prompt, and an image generated from current_prompt, your goal is to refine the current_prompt to better align with the original_prompt while improving clarity and detail. Follow these steps to refine the prompt:

    1. Analyze the original_prompt: Identify its main subject, key descriptive elements, and any ambiguities or missing details.
    2. Evaluate the current_prompt: Compare it to the original_prompt. Determine if it's too short, too long, or misaligned in any way. Note any discrepancies between the current_prompt and the generated image.
    3. Assess the generated image: Describe the visual features of the image (e.g., colors, shapes, textures) and how well they match the current_prompt and original_prompt.
    4. Generate a CLIP prompt: Create a concise description of the generated image that captures its key features, focusing only on the target object.
    5. Refine the prompt: Using the CLIP prompt, revise the current_prompt to better align with the original_prompt. Ensure the refined_prompt follows these rules:
        – Describe only the target object (exclude background, add 'no background').
        – Provide at least 7 values, each with at least 3 words, separated by commas.
        – Add more detail if the current_prompt is too short; simplify if it's too complex, while keeping the main information from the original_prompt.
        – Reduce ambiguity and logical complexity.

    After completing these steps, present your chain of thought reasoning, then output "refined_prompt: object, value1, value2, value3, value4, value5, value6, value7, no background"

Q: 1. original_prompt: "A red rose with petals unfurling in an intricate swirl, each successive layer descending into ever deeper shades of red, the velvety surfaces catching the ephemeral light and casting gentle shadows that reveal the complex network of veins tracing patterns of natural elegance"; 2. current_prompt: "A red rose"; 3. the generated image (shows a simple red rose with no intricate details).

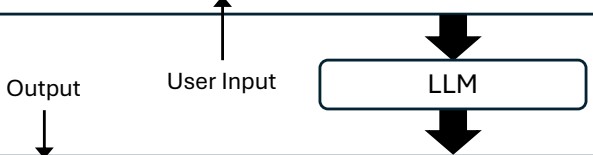

Output      User Input      LLM

A: chain of thought:
    1. Analyze the original_prompt: The main subject is a red rose. Key descriptive elements include "petals unfurling in an intricate swirl," "deeper shades of red," "velvety surfaces," "ephemeral light," "gentle shadows," and "complex network of veins." The prompt is very detailed and poetic, suggesting the user wants a highly specific, elegant image.
    2. Evaluate the current_prompt: "A red rose" is much simpler than the original_prompt. It lacks the swirl, shading, texture, light, shadows, and veins. The generated image likely reflects this simplicity, showing just a basic red rose.
    3. Assess the generated image: The image shows a red rose with smooth petals, uniform red color, and no intricate features, shadows, or veins. The background is minimal.
    4. Generate a CLIP prompt: "Red rose with smooth petals, uniform red color, no intricate details, simple shape, no background."
    5. Refine the prompt: The current_prompt is too vague and short. I need to expand it to include key details from the original_prompt (swirl, shading, texture) while simplifying the language. The refined prompt should focus on the rose and exclude background.

refined_prompt: Rose, petals unfurling swirl, deep red shades, velvety petal texture, gentle light reflection, subtle shadow cast, visible vein network, natural elegance, no background

Figure 6: LLM-CoT-IPR complex prompt.

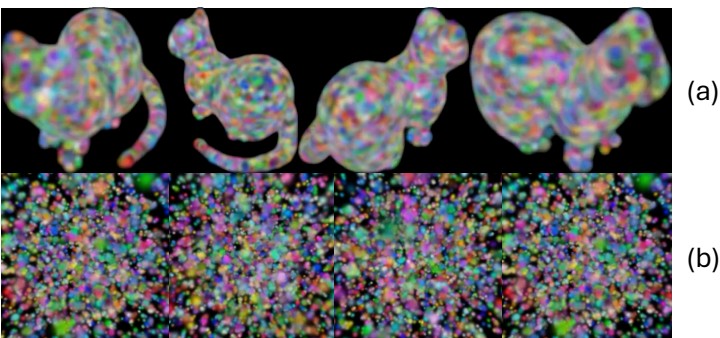

Figure 7: Leveraging the multi-view 3D diffusion prior (MV 3D prior) to initialize the positions of 3D Gaussians at step $t = 0$ enhances geometry's view consistency. (a) With MV 3D prior, it leads to view-consistent initial cat geometry; (b) without MV 3D prior, it only forms scattered 3D Gaussian points. Prompt: *A cat.*

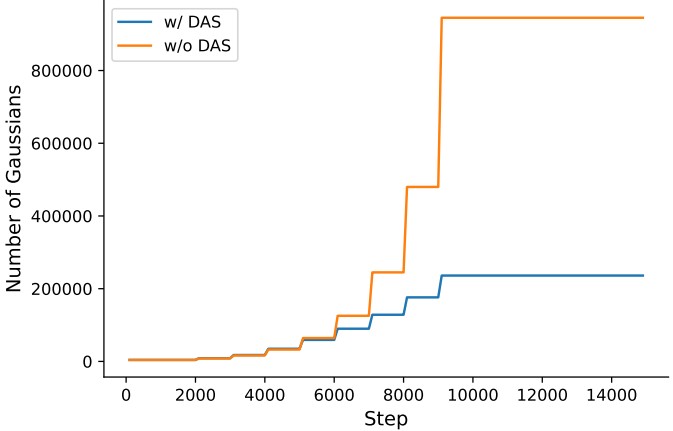

Figure 8: DAS facilitates a more stable increase in the number of Gaussians.

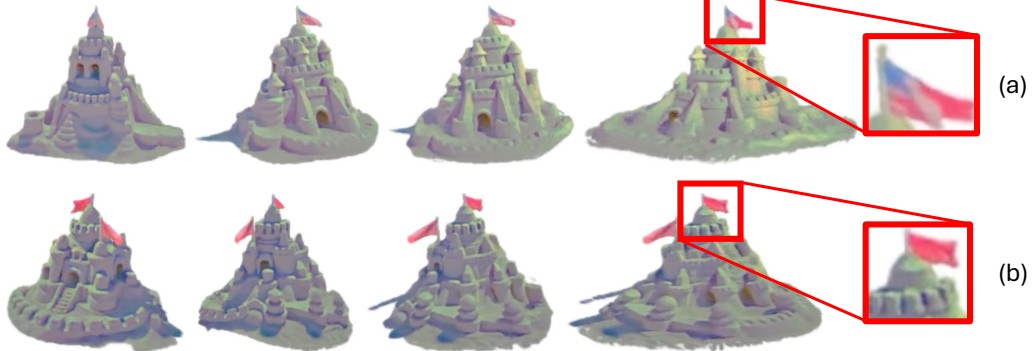

Figure 9: Ablation of multi-view 3D prior in the generated video. (a) Without it, Janus problem occurs; (b) with it, views are consistent.

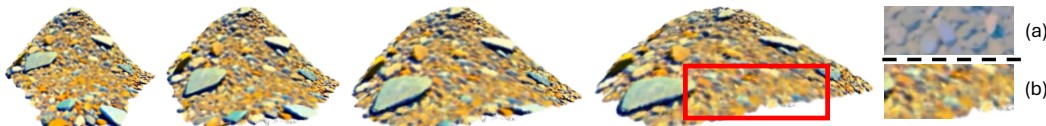

Figure 10: Ablation of DAS in the generated video. (a) With DAS, the result has better details; (b) without DAS, the result has blurry regions.

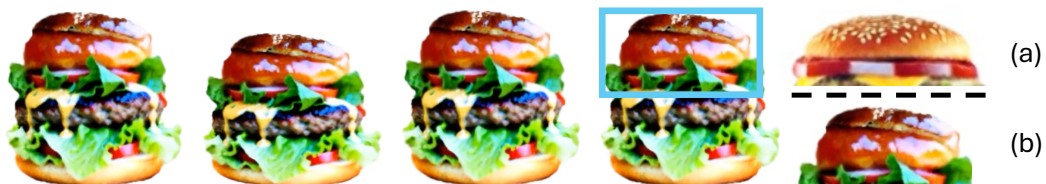

Figure 11: Ablation of LLM-CoT-IPR in the generated video. (a) With LLM-CoT-IPR, burger bread is realistic; (b) without it, the bread is deformed.

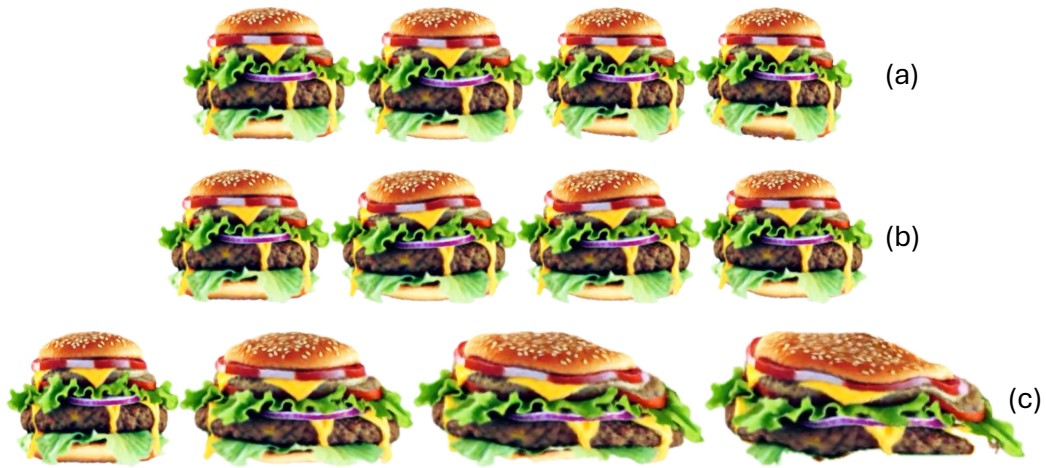

Figure 12: Ablation of 3D-to-motion components in the generated video. (a) Without velocity loss, the rotation is unnatural and velocity is small; (b) without MoEMCMs, the whole burger has distorted motion; (c) without video diffusion prior, the burger melts. Prompt: *A rubber burger falling on a surface.*

