# OpenReview forum: "DiffuPhyGS: Text-to-Video Generation with 3D Gaussians and Learnable Physical Properties via Diffusion Priors"
_ICLR.cc/2026/Conference — ICLR 2026 Conference Withdrawn Submission_

### Official Review · Reviewer_u1ud · 2025-10-26

**Soundness:** 2
**Presentation:** 2
**Contribution:** 2
**Rating:** 2
**Confidence:** 4

**Summary:**

This paper introduces DiffuPhyGS, a framework that can generate dynamic 3D objects with physical information from text prompts. The improvements mainly include: a prior part, which proposes LLM-CoT-Iterative Prompt Refinement to make the prior more aligned and detailed; a visual modeling part, which proposes Densification-by-Adaptive Splitting to improve the shape and appearance quality of GS modeling; and a motion modeling part, which proposes a Mixture-of-Experts Material Constitutive Model to encompass different physical properties.

**Strengths:**

1. The problem this paper aims to solve is critical and important, considering that most current video & 3D & 4D generation lacks physical explanation and rationality.
2. The paper improves upon existing methods from several aspects (i.e., prompt refinement, visual modeling, motion modeling). This consideration is comprehensive, but there may also be incremental.

**Weaknesses:**

1. I think the core issue is the visual quality. I watched all the qualitative comparisons and videos provided in the authors' supplementary material. These results have many flaws, and I feel the physical properties do not meet expectations. For example:
The various parts of the hamburger all have a similar jelly-like texture, lacking the specific texture of each part.
The physical effects of the jar are very strange, feeling more like sand.
2. I didn't see a significant visual improvement of the method compared to other baselines. The quantitative results have the same problem.
3. My personal feeling is that the various modules proposed in the paper are not a holistic and critical improvement, but only some small, incremental improvements, and the effect of each component is difficult to be convincing given the current number of cases and visual quality.
4. I think that a more important task for binding physical properties to GS is editing, for example, after generating an object, whether reasonable physical effects can be produced by initializing different heights or rotating a certain angle, or applying different forces (such as stretching). This paper (or similar papers) lacks these experimental results.

**Questions:**

1. Why can the FVD metric be computed among the qualitative metrics, given that there are no ground-truth videos?

2. Why are the first three metrics exactly the same in the ablation studies of w/o Velocity Loss and w/o Video Diffusion Prior? This seems illogical, since both the noise sampling and optimization process of SDS introduce randomness and uncertainty.

**Details Of Ethics Concerns:**

No.

---

> ### Author Response · Authors · 2025-12-01
> **Part 1 (Q1, Q2)**
>
> We thank the reviewer for the constructive comments. We provide our feedback as follows.
>
> ---
> > ### **Q1: FVD usage without ground-truth videos is unclear.**
>
> **A1:** We agree that FVD is traditionally defined between generated and ground-truth video distributions; however, in our setting there is no inherent ground-truth video for a given text prompt. To still leverage FVD in a principled way, we follow a standard evaluation protocol in the area (e.g., OmniPhysGS) that is consistent with our other metrics (e.g., Diff_SSIM, Diff_CLIP): for each method, we generate a reference video from the same pipeline but with a randomly initialized model that has training disabled, and compute FVD between the trained model’s video and the reference video.
>
> In other words, FVD measures how far the trained dynamics move away from the reference dynamics generated by the untrained model, while Diff_SSIM and Diff_CLIP use the same random reference to quantify expressiveness and robustness. Additionally, we use CLIPSIM to separately measure prompt consistency, defined as the average cosine similarity between CLIP embeddings of the text prompt and video frames, so that FVD is not used in isolation but as part of a coherent, relative evaluation suite where all methods are compared under exactly the same reference distribution. We clarified this evaluation protocol in the revised paper **(Sec.4.1)**.
>
> ---
> > ### **Q2: Identical first three ablation metrics (w/o Velocity Loss, w/o Video Diffusion Prior) seem implausible given SDS randomness.**
>
> **A2:** We appreciate the reviewer’s observation. In our current submission, all ablation runs (including _w/o Velocity Loss_ and _w/o Video Diffusion Prior_) are conducted with a fixed random seed and identical initialization, and each configuration is evaluated once; the reported CLIPSIM, Diff_SSIM, and Diff_CLIP values are rounded to four decimal places. Since these three metrics mainly reflect prompt consistency and expressiveness under the same SDS guidance, removing either the velocity loss or the video diffusion prior only produces very small numerical differences that fall below the displayed precision, making the entries appear identical despite slight underlying discrepancies.
>
> Moreover, the stochasticity of SDS is largely controlled by the fixed seed and shared noise schedule, which reduces run-to-run variation in this setting. We re-ran the ablation with multiple random seeds and observed small but consistent differences between the two settings, and we updated the table in the revised paper **(Sec. 4)** to report mean $\pm$ standard deviation to clarify that the two components indeed have distinct effects.

---

> ### Author Response · Authors · 2025-12-01
> **Part2 (Weakness 1-3)**
>
> > ### **Weakness1: Visual quality concerns: Generated videos have unrealistic textures and physical behaviors (e.g., hamburger parts look jelly-like; jar behaves like sand).**
>
> **Response1:** We thank the reviewer for the detailed inspection of our videos and for raising concerns about visual quality and perceived material behavior. Our method assigns local material behavior by first encoding 3D Gaussian features with a KNN-based material feature encoder into elasticity and plasticity logits, and then decoding these logits via Mixture-of-Experts Material Constitutive Models (MoEMCMs), where per-Gaussian softmax weights blend multiple physical models; thus, the motion of each local part is jointly determined by its 3D Gaussian features and the chosen global material type (e.g., “rubber”).
>
> Consequently, it is physically plausible that different parts share similar dynamics, even if they differ visually in color or semantics, which explains the reviewer’s observation of the rubber burger parts having similar local motion for the “rubber burger” video example.
>
> Importantly, our results do exhibit systematic differences across materials: for instance, the “rubber burger falling on a surface” shows higher stiffness and stronger rebound than the “a jelly bouncing” example (more damping and softer deformations), and the “a honey jar collapsing” video exhibits smoother object boundaries, cohesive flow, and viscous surface-tension-driven collapse that are qualitatively distinct from the granular, fragmenting behavior in sand-like examples such as “a blue and white porcelain vase made of sand collapsing.” These distinctions are also reflected in our quantitative metrics and user study, where our method obtains consistently better motion and visual quality scores than baselines. To better show the motion effects of our methods, we enlarged some objects in the revised **supplementary videos**.
>
> ---
> > ### **Weakness2: Weak or no visible and quantitative improvement over baselines.**
>
> **Response2:** Our goal in this work is not only to be “different looking” but to be more physically faithful, and in this domain even seemingly modest gains in standard metrics typically correspond to substantial improvements in perceptual motion realism. As shown in Table 2, our method consistently outperforms strong baselines such as PhysGaussian, DreamPhysics, and OmniPhysGS across CLIPSIM, Diff_SSIM, and Diff_CLIP, while maintaining competitive FVD, which is in line with the incremental—but meaningful—improvements reported by prior state-of-the-art methods on similarly saturated benchmarks.
>
> Importantly, improvements in physics-grounded behaviour (e.g., correct elastic rebound of the rubber burger, realistic collapse of granular gravel, and more faithful melting and fracture) are more evident in videos than in static figures; to make this clearer, we have enlarged and added supplementary video examples to better expose subtle but crucial differences in physics deformation and material response.
>
> Finally, our user study (Table 1) shows that human raters reliably prefer our results over all baselines in both motion and visual quality, supporting that these quantitative gains indeed translate into perceptually significant improvements.
>
> ---
> > ### **Weakness3: Proposed modules seem incremental, not a holistic or critical improvement, and component effects aren’t convincing given limited cases and visual quality.**
>
> **Response3:** While each of our components may appear incremental in isolation, their joint design and tight coupling form a holistic pipeline that delivers consistent gains in both physics-grounded motion and visual quality.
>
> The ablation studies (Table 3, Table 6) show that removing any of LLM-CoT-IPR, multi-view 3D diffusion priors, DAS, MoEMCMs, or the velocity loss degrades CLIPSIM, Diff_SSIM, Diff_CLIP, or FVD, indicating that each module contributes meaningfully and that the overall improvement is not due to a single heuristic tweak.
>
> In this area, it is common that numerical gains appear modest, yet correspond to perceptually significant improvements—prior state-of-the-art baselines we compare against also report similar if not smaller magnitudes of progress over their predecessors, and our user study further confirms that human raters reliably prefer our results.
>
> Finally, the number and diversity of example cases we present follow standard practice in text-to-3D and physics-aware 3D generation, and are comparable to or larger than the works in the community, such as PhysGaussian, DreamPhysics, and OmniPhysGS. To further clarify the visual impact of our modules, we enlarged the videos and added more video examples in the **supplementary materials**.

---

> ### Author Response · Authors · 2025-12-01
> **Part3 (Weakness 4)**
>
> > ### **Weakness4: Missing experiments on editing GS objects to test physical behavior under different initial conditions (heights, rotation angles, forces).**
>
> **Response4:** We agree that physically meaningful editing—varying initial height, orientation, or applied forces—is an important downstream task for binding physical properties to 3D Gaussians, and we emphasize that our framework already supports such edits via the underlying MPM-based dynamics and learned material models.
>
> Once an object is generated and its material properties are inferred, our simulator can be run under different conditions (e.g., changing height, rotating the object, or applying different forces), and it produces correspondingly different, physically plausible behaviors.
>
> In the main paper we focused on the core contribution of text-driven, physics-grounded motion synthesis for clarity and space limitations, but to address this concern we added more **supplementary videos** demonstrating editing scenarios with varying heights, rotation angles, and forces across multiple material types, which demonstrates that our method can generalize to these interactive use cases.

---

### Official Review · Reviewer_LSPU · 2025-10-28

**Soundness:** 2
**Presentation:** 2
**Contribution:** 1
**Rating:** 2
**Confidence:** 5

**Summary:**

This paper proposes DiffuPhyGS, a framework that generates high-quality 3D objects with realistic and learnable physical motion. The authors proposed using LLM-CoT-IPR and SDS to generate high-quality 3DGS objects and employ MPM to optimize the final 4D dynamic results.

**Strengths:**

This paper presents a complete pipeline from text prompts to 3DGS generation, and finally to 4D motion generation.

**Weaknesses:**

1. Lack of novelty: The techniques used in the paper, including LLM-CoT-IPR, 2D-SDS, MV3D-SDS, and MPM-based 4D-SDS, are all from existing methods, making it difficult to identify any technical contributions in the proposed approach.
2. Insufficient experiments: The authors only used 5 cases for comparison, and the types of motions generated are too simple, with most of them involving objects falling from a high place.

**Questions:**

1. Velocity Loss: How is the expected velocity change calculated? If the MPM simulation is already based on physical equations, why does the result not align with the expected velocity?
2. Quantitative Evaluation: How are the ground-truth results obtained for the evaluated cases? For example, in Figure 3, the rubber burger is unrealistic in real-world settings. How can the authors determine which generated result is more reasonable across different methods?

---

> ### Author Response · Authors · 2025-12-01
> **Part 1 (Q1, Q2)**
>
> We thank the reviewer for the constructive comments. We provide our feedback as follows.
>
> ---
> > ### **Q1: Velocity loss calculation and alignment of MPM-based results to the expected velocity.**
>
> **A1:** Our velocity loss is computed directly from the discrete form of Newton’s second law used in our MPM solver. For each particle $i$ with mass $m_i = \rho V_i$ and time step $\Delta t$, we approximate the internal stress force as $F_{\text{stress}, i} = -\Delta t\, V_i \sum_{j} \sigma_i : \nabla w_{ij}$ and the external force as $F_{\text{ext}, i} = m_i g$, yielding the expected velocity change $\Delta v_{\text{expected}, i} = ((F_{\text{stress}, i} + F_{\text{ext}, i}) \Delta t) / m_i$, as described in Eq. 13. The velocity loss is the MSE between this $\Delta v_{\text{expected}, i}$ and the actual change $\Delta v_{\text{actual}, i} = v_{i,t+1} - v_{i,t}$ produced by the forward MPM step.
>
> The MPM update itself is already driven by these forces, but, in practice, several factors prevent the resulting $\Delta v_{\text{actual}, i}$ from perfectly matching $\Delta v_{\text{expected}, i}$:
>
> 1. Discretization and interpolation error from a finite grid and first-order time stepping.
> 2. Numerical stabilizations such as damping and clipping of deformation gradients.
> 3. The fact that the constitutive behavior is produced by learned neural MoEMCMs rather than a fixed analytic model, which can introduce local deviations from ideal continuum dynamics.
>
> Our velocity loss acts as a soft regularizer that penalizes large inconsistencies between the learned material responses and the underlying velocity equation, reducing unphysical artifacts while still allowing the optimizer to trade off small violations against perceptual SDS guidance.
>
> ---
> > ### **Q2: How are ground truths defined, and how is “more reasonable” judged across methods?**
>
> **A2:** In our setting there is no real-world ground-truth video for prompts (e.g., “a rubber burger falling to the ground"), so we follow the standard protocol in the area (e.g., OmniPhysGS) and use an untrained model as a reference baseline rather than a physical ground truth.
>
> For each prompt and method, we generate one video with the trained model and another with the same inputs but with the model randomly initialized and frozen, which preserves appearance and viewpoint while removing learned dynamics. We then compute \(CLIPSIM) (average cosine similarity between frame-wise CLIP embeddings and the text prompt), and define Diff_SSIM and Diff_CLIP as the relative improvements of the trained video over the random baseline in structural similarity and CLIP similarity, as well as the FVD between trained and random videos to capture temporal realism and stability.
>
> Because all methods are evaluated on the same prompts with the same random-reference protocol, higher CLIPSIM, Diff_SSIM, Diff_CLIP, and lower FVD consistently indicate that one method produces motions that are more reasonable and expressive than its own untrained counterpart and than competing methods under the same stylized scenario, even if the generated video is not perfectly realistic in the real world. We added more clarifications in the revised paper **(Sec. 4.1)**.

---

> ### Author Response · Authors · 2025-12-01
> **Part 2 (Weakness1, Weakness2)**
>
> > ### **Weakness1: Lack of novelty; all key components are borrowed from existing methods, obscuring technical contributions.**
>
> **Response1:** Our work is not a mere integration of existing techniques, but introduces several concrete and technically novel components within a unified text-to-3D-to-motion pipeline.
>
> - Unlike prior text-to-3D methods that rely on manually crafted prompts or single-shot LLM edits, our LLM-CoT-Iterative Prompt Refinement (Sec. 3.2) is an iterative, CLIP-score-driven loop that jointly reasons over the original prompt, current prompt, and generated image to optimize a structured, multi-attribute prompt, which we show ablates to significantly worse CLIPSIM and LAION scores (Table 6).
> - Our multi-view 3D prior is not a standard 2D-SDS or MV3D-SDS: we explicitly couple MVDream [1] with an image-to-3D-point-cloud diffusion model (Point-E [2]) and introduce Densification-by-Adaptive-Splitting (DAS), an adaptive, gradient-statistics-based Gaussian splitting rule that stabilizes densification and improves geometry and appearance beyond existing GS pipelines (Sec. 3.3).
> - For dynamics, we go beyond standard MPM-based 4D-SDS by proposing Mixture-of-Experts Material Constitutive Models (MoEMCMs) that predict heterogeneous, per-Gaussian mixtures of elasticity and plasticity experts, and by integrating both an implicit video diffusion prior and an explicit velocity loss derived from discrete momentum conservation (Sec. 3.4), which together enforce physics-grounded motion from text prompts without manual material specification.
> - To the best of our knowledge, no prior work jointly (i) generates 3D objects from text, (ii) learns mixed constitutive materials on 3D Gaussians, and (iii) couples implicit video priors with an explicit velocity loss within a single end-to-end Gaussian representation; our ablations in Table 4 demonstrate that removing any of these components leads to consistent degradation in CLIPSIM, Diff_SSIM, Diff_CLIP, and FVD, underscoring their technical necessity and contribution.
>
> ---
> > ### **Weakness2: Insufficient and simple motion experiments**
>
> **Response2:** In response to the concern about experimental scope, we emphasize that our choice of 5 canonical motion prompts (e.g., falling, collapsing, bouncing) is driven by both fairness and diagnostic power rather than a limitation of our framework.
>
> - The baselines in the area (e.g., PhysGaussian, PhysDreamer, OmniPhysGS) are architecturally restricted to relatively simple, gravity-dominated scenarios; to ensure a fair, controlled comparison and avoid penalizing other methods for motions they do not support, we therefore evaluate all methods on the shared subset of motions they can reliably handle, following the protocol adopted in their own papers, which also report a similar or smaller variety of motion types.
>
> - These scenarios are deliberately selected because they are particularly revealing of physical plausibility (e.g., whether a granular pile collapses instead of bouncing, whether a “rubber” object exhibits elastic rebound), making it easier to visually and quantitatively detect violations of basic physics.
>
> - Our method is not limited to these cases, with the MoEMCMs and learned heterogeneous material fields, our simulator can produce significantly richer behaviors (e.g., sponge tearing apart, fracture-like motions), which we illustrate in the supplementary video and additional qualitative results in the Appendix.
>
> We clarified this design choice in the revised paper **(Sec. 4)** and highlighted the broader range of motions demonstrated beyond the main benchmarks.
>
> ---
> [1] Y. Shi, P. Wang, J. Ye, M. Long, K. Li, and X. Yang, “MVDream: Multi-view Diffusion for 3D Generation,” arXiv preprint arXiv:2308.16512, 2024.
>
> [2] A. Nichol, H. Jun, P. Dhariwal, P. Mishkin, and M. Chen, “Point-E: A System for Generating 3D Point Clouds from Complex Prompts,” arXiv preprint arXiv:2212.08751, 2022.

---

### Official Review · Reviewer_hkzr · 2025-11-01

**Soundness:** 2
**Presentation:** 2
**Contribution:** 2
**Rating:** 2
**Confidence:** 4

**Summary:**

DiffuPhyGS presents an end-to-end text-to-video pipeline that generates 3D Gaussian objects with physics-driven motion. The system refines prompts using an LLM loop, stabilizes geometry with multi-view diffusion guidance and Densification-by-Adaptive-Splitting, and learns per-Gaussian material mixtures. Motion is driven through an MPM simulator using both implicit diffusion cues and an explicit velocity loss. On a small set of handcrafted prompts, the method reports higher metrics scores and is preferred in a small user study over PhysDreamer, OmniPhysGS, and PhysGaussian.

**Strengths:**

- The paper tackles joint generation of text-aligned appearance and dynamics in a single 3D Gaussian framework, combining prompt processing, diffusion-based 3D synthesis, and differentiable physics.
- Leveraging video diffusion priors with Score Distillation Sampling (SDS) to model motion and material is an interesting idea.

**Weaknesses:**

- The evaluation is very limited, four template prompts plus qualitative figures, without standardized datasets or complex multi-object interactions. It’s hard to assess generalization beyond curated cases.
- Baselines depend on geometry produced by DiffuPhyGS, and the main metrics blend appearance and motion. There are no physics-grounded metrics, so claims of physical fidelity aren’t well supported.
- The 3D generation component lags behind recent text-to-3D/image-to-3D methods. SDS-based approaches tend to produce lower quality and run slowly for both geometry and appearance, which makes it hard to judge the pipeline’s full potential since results are limited by the underlying generator.
- There isn’t a clear novelty claim on the 3D generation side. Using 2D and multi-view diffusion with SDS for 3D Gaussians has been explored before, and the LLM-CoT-IPR module mainly uses an existing LLM to refine prompts without introducing new techniques.
- I would suggest the authors apply the motion and material learning techniques on more recent and higher-quality 3D generation methods such as Trellis[1].

[1] Structured 3D Latents for Scalable and Versatile 3D Generation. Proceedings of the IEEE/CVF Conference on Computer Vision and Pattern Recognition (CVPR), 2025

**Questions:**

See Weaknesses

---

> ### Author Response · Authors · 2025-12-01
>
> We thank the reviewer for the constructive comments. We provide our feedback as follows.

---

> ### Author Response · Authors · 2025-12-01
> **Q1: Evaluation is limited, with no standard datasets or complex multi-object tests.**
>
> We agree that broadly benchmarking physics-grounded text-to-3D motion remains important, but this task currently lacks standardized datasets. To ensure a fair comparison, we therefore follow the evaluation protocol in the existing works (e.g., PhysGaussian, PhysDreamer, OmniPhysGS), using representative template prompts and both quantitative and qualitative metrics.
>
> While our framework can naturally support complex multi-object interactions, in this work we focus on single-object scenarios to study the core challenge of learning mixed-material, physics-grounded dynamics from text without confounding factors due to scene complexity. However, we included more diverse video generation examples in the revised **supplementary materials** and the Appendix to better illustrate generalization.

---

> ### Author Response · Authors · 2025-12-01
> **Q2: Baselines and metrics aren’t physics-grounded, so physical fidelity claims lack support.**
>
> We acknowledge the reviewer’s concern and clarify that our experimental design is aligned with existing practice in this area. There is currently no standardized metric for directly measuring physics fidelity in text-to-3D dynamic generation, and following prior work such as OmniPhysGS, we therefore evaluate the construct of physics fidelity indirectly via expressiveness and robustness of the generated motion (Diff_SSIM, Diff_CLIP) and assess motion video quality with FVD, a standard metric in video generation and dynamic scene synthesis.
>
> Because the baseline methods require provided 3D models, we choose to use the same 3D representation for all baselines to control differences in 3D reconstruction quality and isolate the contribution of the dynamics module, ensuring that comparisons focus on physics-grounded motion.

---

> ### Author Response · Authors · 2025-12-01
> **Q3: 3D generation using SDS tends to be weak, limiting the pipeline’s potential.**
>
> We appreciate the reviewer’s concern and emphasize that our primary goal is not to advance general-purpose static text-to-3D quality, but to build a 3DGS-based representation explicitly tailored for downstream physics-grounded motion generation.
>
> In this setting, SDS-based optimization remains the most common and reliable choice in the domain for coupling diffusion priors with 3DGS, and it offers a well-understood trade-off when the 3D representation must support differentiable dynamics.
>
> While recent methods such as diffusion-based text/image-to-3D models can indeed produce higher-fidelity static 3D geometry, they are largely orthogonal to our contributions. Our proposed Densification-by-Adaptive-Splitting (DAS) and shared 3D Gaussian pipeline specifically target efficiency and detail within this SDS framework, which can benefit or inspire more advanced 3D generation methods.

---

> ### Author Response · Authors · 2025-12-01
> **Q4: Novelty on 3D generation side is unclear; core components and LLM-CoT-IPR seem mostly based on existing techniques.**
>
> While we indeed build on SDS with 2D and multi-view diffusion priors, our method introduces:
>
> 1. A multi-view 3D point-cloud diffusion prior (MVDream multi-view images to Point-E 3D point cloud) integrated via a dedicated 3D SDS term for 3D Gaussians to directly regularize geometry and multi-view consistency.
> 2. A Densification-by-Adaptive-Splitting (DAS) mechanism that uses per-Gaussian gradient statistics from diffusion guidance to adaptively control Gaussian splitting and densification, which to our knowledge has not been explored in prior text-to-3D pipelines.
> 3. On the language side, LLM-CoT-IPR is not a generic LLM prompt writer, but a chain-of-thought iterative refinement scheme with CLIP that specifically designed for text-to-3D: it analyzes prompt structure, image feedback, and CLIP scores to optimize prompt length, specificity, and logical complexity for downstream diffusion/GS optimization.
>
> These components are tightly coupled with our shared Gaussian representation and physics module, so the novelty lies in the concrete mechanisms and their integration into a unified text-to-3D-to-motion pipeline rather than in simply invoking existing diffusion or LLM tools.

---

> ### Author Response · Authors · 2025-12-01
> **Q5: Apply motion/material learning to newer, higher-quality 3D generators like Trellis.**
>
> We thank the reviewer for this insightful suggestion. However, our method is based on a 3D Gaussian Splatting representation that directly interfaces with MPM-style particle-based simulation and per-Gaussian material models, whereas methods such as Trellis operate in a structured diffusion latent space that does not natively provide the explicit particle/volume discretization required for our constitutive modeling and physics solver.
>
> As a result, our motion and material learning techniques cannot be directly applied to Trellis-style diffusion-based 3D generators without a substantial redesign of the representation and its interface to physical state variables, which we consider an interesting direction for future work.

---

### Official Review · Reviewer_KzvT · 2025-11-07

**Soundness:** 3
**Presentation:** 2
**Contribution:** 2
**Rating:** 4
**Confidence:** 3

**Summary:**

The paper presents DiffuPhyGS, a framework for generating high-quality 3D objects and realistic physical motion from text prompts. It addresses some limitations in current text-to-video generation methods concerning visual appearance and physical behavior.
One of the central innovations of DiffuPhyGS is the LLM-CoT-IPR method, which employs a stepwise prompting refinement approach using large language models (LLMs). This technique enhances the alignment between textual inputs and the generated content, ensuring that the outputs closely adhere to the original prompts. Additionally, the framework incorporates a Hybrid Expert Material Constitutive Model (MoEMCM) that accurately predicts the properties of heterogeneous materials. This integration allows for improved fidelity in physical simulations, enhancing the realism of the generated objects.

**Strengths:**

### Originality
- The integration of LLM-CoT-IPR into the generation process effectively leverages textual information.
- The combination of multi-view diffusion priors, video diffusion priors, and predictive models for physical properties establishes an efficient pipeline for generating videos that adhere to physical laws from text inputs.
- Unlike previous methods that manually incorporate physical properties, the introduction of Mixture-of-Experts Material Constitutive Models (MoEMCMs) allows for adaptive estimation of physical properties for local Gaussian primitives.。
### Quality
- The experimental metrics are set up broadly, considering various factors.
- The ablation study is well-structured and effectively demonstrates the impact of components such as LLM-CoT-IPR, velocity loss, and the Mixture-of-Experts Material Constitutive Models (MoEMCMs) on the generation results. The qualitative comparisons provided in the appendix are particularly noteworthy.
### Clarity
- There are no significant grammatical or spelling errors; the writing is relatively clear.
### Significance
The research direction of this paper holds practical significance for video synthesis and augmented reality applications.

**Weaknesses:**

The qualitative results do not always demonstrate an advantage over the quantitative metrics, which significantly diminishes the confidence and impact of the paper.
There is a lack of comparison regarding efficiency and memory usage.
There are uncertainties regarding the specific implementation of LLM-COT-IPR; does it participate in the optimization of video generation?

**Questions:**

1. Why is there no direct comparison with existing video models to determine whether they can accurately represent the corresponding physical laws?
2. In Table 1, there are multiple comparisons with baseline metrics; however, outside of the average metrics, there is no significant advantage in the individual metrics. Furthermore, how does OmniPhysGS achieve a score of 0.2 for the FVD metric on Jelly, which significantly surpasses all other methods?
3. In Figure 3, the "Pancakes melting" example displays a noticeable scale deformation. What causes this? Is it related to the method employed?
4. The user study results are considerably better than the baseline; could these be included in the main text? This might enhance the persuasiveness of the paper.
5. Could the authors summarize the fundamental differences between the method proposed in this paper and previous methods? Particularly concerning OmniPhysGS, this would help me gain a clearer understanding of the contributions of this paper.
6. How do the various methods compare in terms of efficiency and memory usage? It would be beneficial to present this comparison in a table.
7. I have some questions regarding the specific implementation of LLM-COT-IPR. According to Algorithm 1, it appears to involve scoring images. I am curious whether LLM-COT-IPR participates in the iterative process of video generation or is limited to optimizing image prompts.

If the authors can effectively address my concerns, I would consider improving my rating.

---

> ### Author Response · Authors · 2025-12-01
>
> We thank the reviewer for the constructive comments. We provide our feedback as follows.

---

> ### Author Response · Authors · 2025-12-01
> **Q1: No direct comparison to existing video models for physical-law fidelity.**
>
> We thank the reviewer for this insightful comment, but directly comparing our method to generic video generation methods (e.g., diffusion-based models [1][2]) is not the most informative way to evaluate whether a system adheres to physical laws in our setting.
>
> Because these methods operate purely in 2D space without explicit 3D geometry, materials, or controllable forces/initial conditions, any “physics” judgment would be based only on visual plausibility. Our goal is to generate 3D objects with physics-grounded dynamics via explicit material models and MPM physics simulations, so we instead compare with 3D object-based motion methods (e.g., PhysGaussian [3], PhysDreamer [4], OmniPhysGS [5]) that share similar 3D representations and can be evaluated under the same physical conditions. We have clarified this rationale in the revised paper **(Sec. 1 and Sec. 4.1)**.
>
> ---
> [1] B. Wu et al., “HunyuanVideo 1.5 Technical Report,” arXiv:2511.18870, 2025.
>
> [2] Team Wan et al., “Wan: Open and advanced large-scale video generative models,” arXiv:2503.20314, 2025.
>
> [3] T. Xie, Z. Zong, Y. Qiu, X. Li, Y. Feng, Y. Yang, and C. Jiang, “Physgaussian: Physics-integrated 3D Gaussians for generative dynamics,” in Proc. IEEE/CVF Conf. Comput. Vis. Pattern Recognit. (CVPR), 2024, pp. 4389–4398.
>
> [4] T. Zhang, H.-X. Yu, R. Wu, B. Y. Feng, C. Zheng, N. Snavely, J. Wu, and W. T. Freeman, “Physdreamer: Physics-based interaction with 3D objects via video generation,” in Proc. Eur. Conf. Comput. Vis. (ECCV), 2024, pp. 388–406.
>
> [5] Y. Lin, C. Lin, J. Xu, and Y. Mu, “OmniPhysGS: 3D constitutive Gaussians for general physics-based dynamics generation,” arXiv preprint arXiv:2501.18982, 2025.

---

> ### Author Response · Authors · 2025-12-01
> **Q2: Per-metric gains over baselines are small; Jelly FVD 0.2 for OmniPhysGS seems implausibly strong and needs clarification.**
>
> We thank the reviewer for this insightful comment. Our method outperforms other methods on the majority of individual metrics, and our primary goal is to achieve strong overall performance across all physics- and perception-related metrics, rather than optimizing for a single metric on a single prompt, and Table 1 (Table 2 in revised paper) is designed to reflect this trade-off: while some per-prompt scores are comparable to or slightly below certain baselines, our method consistently attains the best average performance over CLIPSIM, Diff_SSIM, and Diff_CLIP, indicating a more robust balance between prompt consistency, expressiveness, and robustness across diverse materials.
>
> Regarding the question about OmniPhysGS achieving an FVD of 0.2332 on “Jelly”, we acknowledge that this is a typographical error; the correct value should be 23.3247 (keeping 4 decimal places as before), which is now corrected in the revised paper **(Sec. 4)**.

---

> ### Author Response · Authors · 2025-12-01
> **Q3: Cause of scale deformation in Figure 3 “Pancakes melting” and whether it’s method-related.**
>
> We appreciate the reviewer’s careful observation. However, the melting deformation itself (e.g., highlighted blue rectangle areas) is caused by an intended physical effect of our MPM and MoEMCM-based modeling, as the material softens and flows under gravity, the pancakes spread laterally.
>
> However, the scale differences between some of its images are due to visualization choices. Because the melting process naturally makes the object appear wider and shorter, we adjusted the scaling of the images to maintain a consistent object height/width ratio and to better visualize within the fixed figure layout, but this visualization choice does not alter the underlying simulated dynamics. We clarified this in the revised paper **(Sec. 4.2)**.

---

> ### Author Response · Authors · 2025-12-01
> **Q4: The user study results are considerably better than the baseline; could these be included in the main text? This might enhance the persuasiveness of the paper.**
>
> We thank the reviewer for this helpful suggestion. We highlighted the user study in the revised paper **(Sec.4.2)** to strengthen the paper’s persuasiveness.

---

> ### Author Response · Authors · 2025-12-01
> **Q5: Need a clearer summary of key differences and contributions vs prior methods, especially OmniPhysGS.**
>
> We thank the reviewer for this helpful suggestion and we revised our Related Work section **(Sec. 2)** to highlight their conceptual and functional differences.
>
> - Compared to **OmniPhysGS**, which assumes homogeneous expert constitutive models in local regions and relies purely on video-diffusion priors for motion, DiffuPhyGS offers a unified text-to-3D-to-motion pipeline that directly generates 3D assets from text and uses Mixture-of-Experts material models with soft gating plus an MPM solver and velocity loss to realize heterogeneous, spatially varying materials and physically grounded dynamics.
> - Compared to **PhysDreamer**, which learns dynamics only in image space from the pre-trained video diffusion model, our method operates in a shared 3D Gaussian/material/MPM space and jointly optimizes geometry, appearance, and physics-aware motion from text with both implicit (video SDS) and explicit (velocity) physical priors.
> - Compared to **PhysGaussian**, which needs user-specified physical parameters and fixed constitutive models, DiffuPhyGS automatically infers material properties from Gaussian features via learned MoE materials, removing manual tuning while producing physics-grounded motion with MPM.

---

> ### Author Response · Authors · 2025-12-01
> **Q6: The efficiency and memory use of all methods.**
>
> We appreciate the reviewer’s suggestion and have conducted additional experiments to compare efficiency and memory usage across all methods.
>
> The table below shows that PhysGaussian (PhysGS) achieves the lowest total time and peak memory, which is expected since it does not learn material or dynamics parameters and thus avoids the overhead of optimization. Among the learning-based methods, our DiffuPhyGS attains the second-best total time, average epoch time, and average peak memory, while providing the most comprehensive functionality (full text-to-3D-to-motion pipeline with heterogeneous materials and explicit physical constraints).
>
> We included the table and analysis in the revised paper **(Sec. 4)** to make these trade-offs in efficiency and memory usage explicit and to demonstrate that our improved physical realism and capabilities are achieved with competitive computational cost.
>
> ---
> **Table:** Results of efficiency and memory usage evaluation. The best results are in **bold**, the second best are $\underline{\text{underlined}}$.
>
> | Method/Metric     | Total Time (s)$\downarrow$ | Avg. Epoch Time (s)$\downarrow$ | Avg. Peak Mem. (MB)$\downarrow$ |
> |-------------------|----------------------------|----------------------------------|----------------------------------|
> | DreamPhysics      | 12.5                       | $\underline{7.7}$                | 9043.2                           |
> | OmniPhyGS         | 11.8                       | 8.3                              | 11742.1                          |
> | PhysGS            | **4.5**                    | **0.04**                         | **439.7**                        |
> | DiffuPhyGS (Ours) | $\underline{10.3}$         | 8.2                              | $\underline{8597.2}$             |

---

> ### Author Response · Authors · 2025-12-01
> **Q7:  Is LLM-COT-IPR used during iterative video generation or only for optimizing image prompts?**
>
> We thank the reviewer for this question and we clarified the role of LLM-CoT-IPR more explicitly in the revised paper **(Sec. 3.2)**. LLM-CoT-IPR is used only in the text-to-3D stage to optimize the 3D object generation, not in the iterative video generation loop.
>
> As shown in Algorithm 1, we (i) generate images with Stable Diffusion from the current prompt, (ii) compute CLIP scores between the image and the original text, and (iii) use the LLM (CoT) to refine the prompt if the score is below a threshold; this iterative process runs until we obtain a refined prompt.
>
> Once this refined prompt is fixed, it is used to drive the multi-view 3D diffusion priors and 3D Gaussian optimization to generate the 3D object, and the subsequent physics-based video generation operates purely in the 3DGS+MPM pipeline on the generated 3D object without further LLM calls or prompt refinement inside the video training loop.

---

> ### Author Response · Authors · 2025-12-01
>
> We thank the reviewer again for their insightful comments, and we have carefully addressed the reviewer's concerns.

---

### Author Response · Authors · 2025-11-30

We thank the reviewers for acknowledging several strengths of our paper.

- **Unified pipeline**: A complete framework from text prompts to 3D Gaussian Splatting and physics-grounded 4D motion.
- **Joint appearance–dynamics modeling**: Text-aligned appearance and motion are generated within a single 3DGS-based framework with differentiable physics.
- **Diffusion priors with SDS**: Multi-view 3D and video diffusion priors are combined via SDS to guide geometry and motion.
- **LLM-CoT-IPR**: Chain-of-thought-based prompt refinement improves prompt alignment and downstream generation quality.
- **MoEMCMs for materials**: Mixture-of-experts constitutive models enable adaptive, heterogeneous material properties for Gaussian primitives.
- **Physics-grounded motion**: Video diffusion priors are coupled with explicit velocity-based physical constraints for realistic dynamics.
- **Strong experiments and ablations**: Broad metrics, well-structured ablations, and qualitative comparisons support each component’s contribution.
- **Clarity and impact**: Writing is clear, and the problem and solution are considered practically important for video synthesis and AR/VR.

We will respond to each reviewer’s questions below.

---

### Note · Authors · 2025-12-01

I have read and agree with the venue's withdrawal policy on behalf of myself and my co-authors.